# Subfunctionalisation and self-repression of duplicated *E1* homologues finetunes soybean flowering and adaptation

Chao Fang[1,2,3,5], Zhihui Sun[1,5], Shichen Li[1,5], Tong Su[1,5], Lingshuang Wang[1,5], Lidong Dong [1,5], Haiyang Li[1], Lanxin Li[1], Lingping Kong [1], Zhiquan Yang [1], Xiaoya Lin[1], Alibek Zatybekov[4], Baohui Liu [1,2] ✉, Fanjiang Kong [1,2,3] ✉ & Sijia Lu [1] ✉

Soybean is a photoperiod-sensitive staple crop. Its photoperiodic flowering has major consequences for latitudinal adaptation and grain yield. Here, we identify and characterise a flowering locus named Time of flower 4b (*Tof4b*), which encodes E1-Like b (E1Lb), a homologue of the key soybean floral repressor E1. Tof4b protein physically associates with the promoters of two *FLOWERING LOCUS T* (*FT*) genes to repress their transcription and delay flowering to impart soybean adaptation to high latitudes. Three E1 homologues undergo subfunctionalisation and show differential subcellular localisation. Moreover, they all possess self-repression capability and each suppresses the two homologous counterparts. Subfunctionalisation and the transcriptional regulation of *E1* genes collectively finetune flowering time and high-latitude adaptation in soybean. We propose a model for the functional fate of the three *E1* genes after the soybean whole-genome duplication events, refine the molecular mechanisms underlying high-latitude adaption, and provide a potential molecular-breeding resource.

Soybean (*Glycine max* (L.) Merr.) is a major source of vegetable oil and the largest source of protein for animal feed[1,2]. As a short-day plant, soybean is extremely sensitive to photoperiod, such that its photoperiodic response and induction of flowering greatly influence its latitudinal adaptation and grain yield[3]. It is widely believed that modern cultivated soybean was domesticated from wild soybean (*Glycine soja*) in the Huanghuai region in Central China around 5000 years ago[4]. Reduced photoperiod sensitivity and early flowering were target traits during domestication, which enabled soybean to adapt to higher latitudes. The underlying genetic mechanism was the stepwise selection for loss-of-function alleles of two

*PSEUDO-RESPONSE-REGULATOR* (*PRR*) genes, *Tof11* and *Tof12*[5]. Following domestication, photoperiodic flowering time remains a major target of artificial selection during soybean improvement, thus enabling the gradual extension of soybean cultivation to a wide latitudinal range from 53° N to 35° S[6,7].

During the expansion of cultivated soybean around the world, components of the legume-specific *E1*-family photoperiodic flowering pathway accumulated polymorphisms[3,6,8–19]. Among this unique flowering pathway, *E1* (*Glyma.06G207800*) is a core component that encodes a transcription factor containing a B3 domain and a putative bipartite nuclear-localisation signal (NLS)[20]. *E1* has four major recessive

[1]Guangdong Provincial Key Laboratory of Plant Adaptation and Molecular Design, Innovative Center of Molecular Genetics and Evolution, School of Life Sciences, Guangzhou University, Guangzhou Higher Education Mega Center, Guangzhou, China. [2]The Innovative Academy of Seed Design, Key Laboratory of Soybean Molecular Design Breeding, Northeast Institute of Geography and Agroecology, Chinese Academy of Sciences, Harbin, China. [3]College of Agronomy and Biotechnology, China Agricultural University, Beijing, China. [4]Laboratory of Molecular Genetics, Institute of Plant Biology and Biotechnology, Almaty, Kazakhstan. [5]These authors contributed equally: Chao Fang, Zhihui Sun, Shichen Li, Tong Su, Lingshuang Wang, Lidong Dong. ✉e-mail: liubh@gzhu.edu.cn; kongfj@gzhu.edu.cn; lusijia@gzhu.edu.cn

alleles: *e1-as* (a single missense point mutation), *e1-fs* (a 1-bp deletion leading to frame-shift), *e1-nl* (-130 kb deletion comprising the *E1* gene) and *e1-b3a* (three SNPs and a 2-bp deletion in the middle of the B3 domain)[21]. Due to two rounds of whole-genome duplication in soybean, *E1* has two homologues, *E1 LIKE a* (*E1La, Glyma.04G156400*) and *E1 LIKE b* (*E1Lb, Glyma.04G143300*)[22]. However, the functional and evolutionary relationships amongst the three soybean *E1* homologues are poorly characterised.

The impairments in the *E1* family itself and its positive regulators, such as *E3* and *E4* (two *PHYTOCHROME A* homologues), *E2* (*GIGANTEA*), *Tof11/Tof12* (*PRR3a/b*), *FKFs* (*Flavin-binding, Kelch repeat, F-box*) contributed to cultivated soybean adapting to high latitudes[5,20,23–26]. However, the genetic basis for high-latitude adaption for wild soybean accessions remain poorly understood. So far, only three genes have been reported to contribute to the expansion of wild soybean to higher latitudes, namely an enhancer of *E1* family (*E3*), an *E1* family member (*E1La*) and an element regulated by the *E1* family (*Tof5/FRUITFULL*)[8,27]. The genetic basis behind wild soybean adapting to high latitudes merits further investigation.

In this study, we identify a flowering-time locus that we name Tof4b and confirm that its causal gene is *E1Lb*, a homologue of *E1*. We provide mechanistic insight into how *Tof4b* affects soybean flowering time, trace the evolutionary trajectory of *Tof4b* and explore its role in high-latitude adaptation of both cultivated and wild soybean. The *E1* genes all possess the ability for self-repression and suppression of homologous gene and have undergone subfunctionalisation during evolution to form a rare genetic module participating in the photoperiodic flowering pathway. These results refine the molecular mechanisms underlying high-latitude adaption of soybean.

## Results

### Identification of the *Tof4b* locus

In a QTL mapping study using a population developed from a cross between Dongnong 50 (DN50) and Williams 82 (Wm82), a QTL named *Tof4* was initially mapped to an interval at a physical position of 15,526,145–44,508,948 bp on chromosome 4[28]. *E1La* was identified as a candidate gene[8]. However, *Tof4/E1La* alone cannot fully explain the flowering-time differences caused by this chromosomal region[28]. Further inspection revealed a secondary population with homozygosity at the interval including *E1La*, but heterozygosity for the interval between 15,526,145–29,675,002 bp showed a difference in flowering time. These observations suggest the presence of another locus (*Tof4b*). We generated a large (n = 5764) inbred $F_8$ segregating population from the cross between DN50 and Wm82 by continually selecting for a residual-heterozygous line for *Tof4b*. Because this interval is located at the centromere, we could only refine the *Tof4b* locus within the 21,162,447–29,675,002-bp interval on chromosome 4 (Fig. 1a; Supplementary Data 1). This region harbours a member of the *E1* family, *E1Lb* (*Glyma.04G143300*), a floral repressor controlling night-break responses that is associated with late flowering under non-inductive long-day photoperiods[26,29]. The *E1Lb* coding sequence was identical in Wm82 and DN50, but three variants were found in the *E1Lb* promoter of the parents: a 50-bp deletion and two SNPs in DN50 (Fig. 1b). The 50-bp deletion harbours a Box-4 element, a light-responsive element that plays a role in light-controlled transcriptional activity[30]. Expression of *E1Lb* in NILs of *Tof4b* from Wm82 was statistically significantly higher than from DN50 (Fig. 1c), implying that the effect of *Tof4b* might be caused by differential promoter activity. To determine which promoter polymorphism/s exerts this, we designed four different reporter-gene vectors and found that the 50-bp deletion, but not the other SNPs, decreased the activity of the *E1Lb* promoter (Fig. 1d, e and Supplementary Fig. 1). We therefore concluded *E1Lb* underlies *Tof4b* and the 50-bp promoter deletion is the causative mutation.

### *Tof4* encodes an E1Lb protein

To investigate *Tof4b* function in soybean, we generated *tof4b* null mutants by CRISPR–Cas9 gene editing in both Wm82 and DN50 backgrounds, designated *tof4b^CR-DN50* and *tof4b^CR-Wm82*. DNA sequencing identified 2-bp and 4-bp deletions in the *E1Lb* coding sequence in the Wm82 and DN50 mutants, respectively. The two deletions resulted in frameshifts at 369 bp and 370 bp in the 579-nucleotide coding sequence, respectively. (Supplementary Fig. 2a, b). Both *tof4b^CR* lines flowered early under long-day photoperiods relative to their respective wild types (Fig. 1f–i). We developed stable-transgenic lines in the Wm82 background over-expressing the *Tof4b* coding sequence fused to a triple FLAG tag (*Tof4b–3xFLAG*, designated *Tof4b-OE*) (Supplementary Fig. 3a). Two genetically independent T3 homozygous lines had delayed flowering under long-day photoperiods relative to Wm82 controls (Supplementary Fig. 3b, c). Taken together, these data confirmed that *E1Lb* functions as a flowering inhibitor and is the causative gene underlying the *Tof4b* locus.

### *Tof4b* delays soybean flowering under long days by repressing *FT2a* and *FT5a*

*E2*, *E3* and *E4* are upstream regulators of *E1* in the *E1*-dependent soybean photoperiodic pathway and play key roles in adaptation to high latitudes[10,14,31,32]. To understand their influence on *Tof4b*, we compared *Tof4b* expression in near-isogenic line (NIL) pairs for *E2* and *E3/E4*. *Tof4b* expression was higher in NIL-*E3/E4* compared with in NIL-*e3/e4* under long-day photoperiods (Fig. 2a), suggesting that these two *PHYTOCHROME A* (*PHYA*) homologues, *E3* (*PHYA3*) and *E4* (*PHYA2*) up-regulate the expression of *Tof4b*, consistent with a previous study[31,33]. In addition, *Tof4b* is up-regulated in NIL-*E2* plants versus NIL-*e2* plants, indicating that *E2* also promotes the expression of *Tof4b* (Fig. 2b).

Virus-induced silencing of *E1Lb* up-regulated the expression of *FT2a* and *FT5a*[29]—the two major soybean florigens. Here, we observed a similar trend in that *FT2a* and *FT5a* were up-regulated in wild-type Wm82 relative to *tof4b^CR-Wm82*, confirming that *E1Lb* is a repressor of *FT2a* and *FT5a* (Fig. 2c, d). To determine whether E1lb protein directly binds to the promoters of *FT2a* and *FT5a*, ChIP–qPCR assays using the *Tof4b-OE* transgenic lines and wild-type Wm82 were performed. Like its homologues E1 and E1La[8], Tof4b–3xFLAG directly binds to the P2 and P4 fragments in the *FT2a* promoter and the P1 and P4 fragments in the *FT5a* promoter (Fig. 2e, f). These results suggested that Tof4b–3xFLAG delays flowering by directly binding to the promoters of *FT2a* and *FT5a* to suppress their expression. Taken together, *Tof4b* encodes a *E1*-family protein, E1Lb and *Tof4b* have a similar genetic mechanism to its homologues: it is enhanced by E2 and E3/E4 and directly suppresses the expression of *FT2a* and *FT5a* to delay soybean flowering under long-day conditions.

### Soybean E1-family members have subfunctionalised

To further understand the relationship amongst *E1*, *E1La* (*Tof4*) and *E1Lb* (*Tof4b*), we scored the flowering times of a series of mutants in the Wm82 background described previously: *e1 E1La E1Lb* (a null *e1* mutant), *e1-as e1la E1Lb*, *e1-as E1La e1lb*, *e1-as e1la e1lb* and *e1 e1la e1lb*[31] grown under long-day conditions. The Wm82 background is designated as '*e1-as E1La E1Lb*' and carries a weak *e1* mutant allele previously designated '*e1-as*' containing an amino-acid substitution in the nuclear-localisation sequence (NLS)[19]. *e1* mutants generated here are null mutants (Supplementary Fig. 4). Although *e1-as* (Wm82) is a partial loss-of-function allele, the difference in days to flowering between *e1-as E1La E1Lb* and *e1 E1La E1Lb* (14 days) is much greater than that between *e1-as E1La E1Lb* and *e1-as e1la E1Lb* (4 days), *e1-as E1La E1Lb* and *e1-as E1La e1lb* (4.6 days), respectively. Moreover, the flowering-time difference between *E1La E1Lb* and *e1la e1lb* also depended on the strength of the *E1* allele: 17.8 days and 14.3 days for *e1-as* and *e1*, respectively. These results suggested that *E1* has a much stronger effect on delaying flowering than either *E1La* or *E1Lb*, and there might

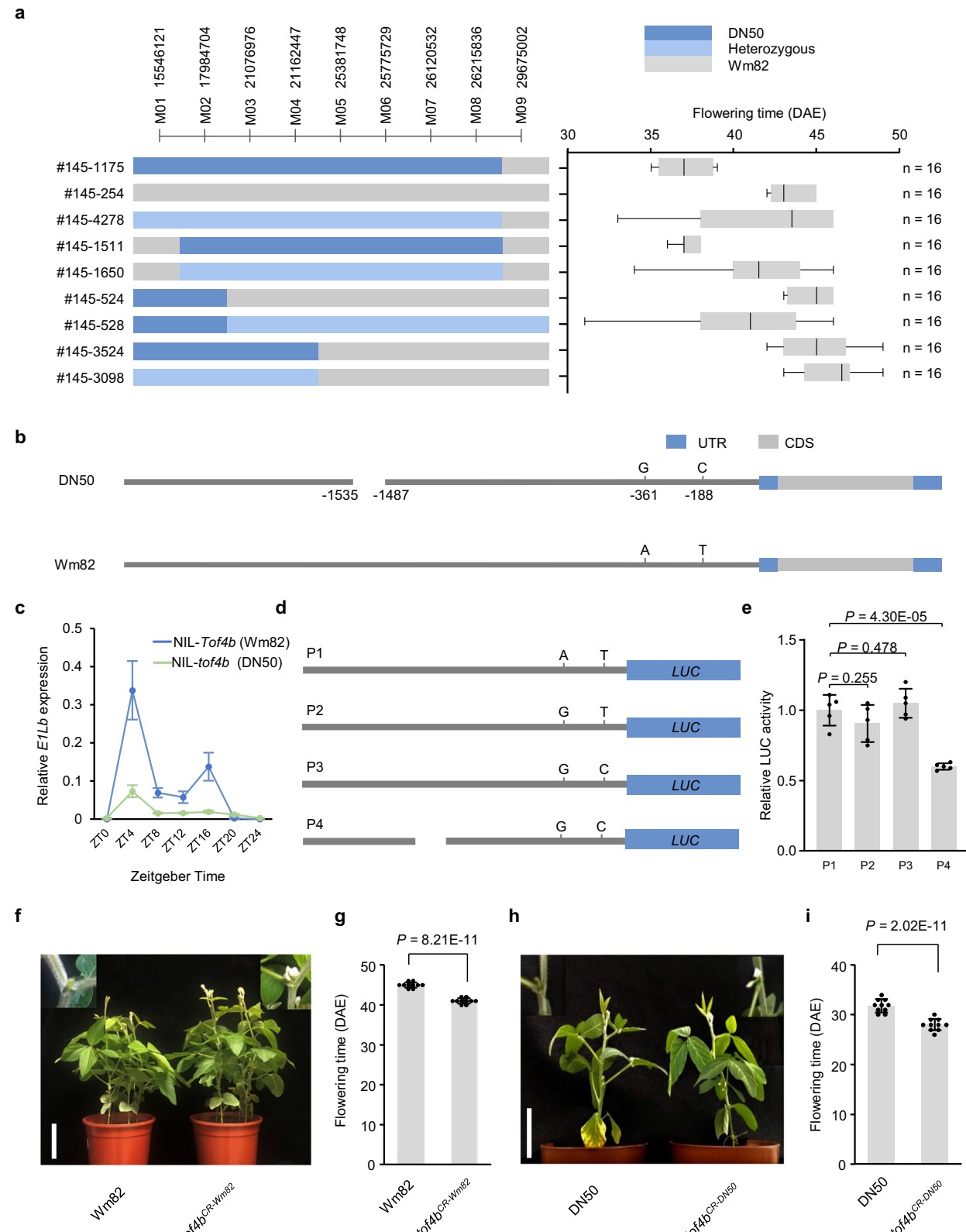

be a regulatory relationship between *E1* and two *E1-LIKE* genes. To explain the relationship between the three *E1* genes, we characterised them at the protein and transcript levels as follows.

*E1* encodes a 174-amino acid (aa) protein, while *E1La* and *E1Lb* both encode 192-aa proteins[20,29]. We found a potential translation-initiation site upstream of the previously annotated translation initiation site, extending the length of E1 to 193-aa, similar to E1La and E1Lb

(Supplementary Fig. 5a). By individually replacing the *E1* ATG start codons with ATT codons, both proteins are indeed expressed and do not markedly affect their corresponding nuclear-localisation (Supplementary Fig. 5b–d). Using transient-reporter assays, we found that the longer E1 protein has much stronger transcriptional inhibition on the *FT2a* promoter than the previously characterised E1, suggesting that the longer E1 isoform possesses greater biological activity

**Fig. 1 | Identification of *Tof4b* and the validation of *E1Lb* underlying *Tof4b*.**
**a** Delimitation of the *Tof4b* locus to an 8512-kb region on chromosome 4 in a segregating, heterozygous inbred family from a cross between Williams 82 (Wm82) and Dongnong 50 (DN50). Flowering time was recorded in 2020 in Harbin under natural long-day conditions. Segregation of flowering time is shown in box-plot format, where the interquartile region, median and range are represented by the box, the bold vertical line and the horizontal line, respectively. n represents the number of plants. **b** Schematic of allelic variation in the *Tof4b* candidate gene *E1Lb* in Wm82 and DN50. Numbers refer to the nucleotide-residue position relative to the annotated translational start site. **c** Expression of *E1Lb* in the *Tof4b* NILs (Near Isogenic Lines). Data are the mean ± S.D. of n = 3 biological replicates and relative expression was normalised to *Tubulin* (*Glyma.05G157300*). ZT: Zeitgeber time. **d** Schematic of the constructs used for a transient reporter-gene assay to validate

the causal mutations of *E1Lb*. P1 represents the promoter of Wm82, P4 represents the promoter of DN50, P2 and P3 are the transitional form. **e** Relative LUC activity driven by the promoters in (**d**). Data are the mean ± S.D. of n = 5 biological replicates. The two-sided Student's *t*-test was used to generate the *P* values. **f** Phenotypes of wild-type c.v. Wm82 and *tof4b*^CR-Wm82 plants cultivated under long-day conditions in a growth chamber. Scale bar = 10 cm. **g** Flowering time of Wm82 and *tof4b*^CR-Wm82 plants grown as described in (**f**). Data are the mean ± S.D. of n = 10 plants. DAE Days after emergence. The two-sided Student's *t*-test was used to generate the *P* values. **h** Phenotypes of wild-type c.v. DN50 and *tof4b*^CR-DN50 plants cultivated under long-day conditions in a growth chamber. Scale bar = 10 cm. **i** Flowering time of DN50 and *tof4b*^CR-DN50 plants grown as described in (**h**). Data are the mean ± S.D. of n = 10 plants. The two-sided Student's *t*-test was used to generate the *P* values. Source data are provided as a Source Data file.

(Supplementary Fig. 5e). In subsequent experiments, we use the longer 193-aa E1 isoform.

An alignment of the three E1 amino-acid sequences found four substitutions (position 7 S→I, position 12 T→K, position 13 T→I and position 14 L→I) among E1 and E1La/E1Lb in the N-terminus near the putative bipartiteNLS[20] (Fig. 3a). To test whether these substitutions affect the nuclear targeting of two E1-Like proteins, we developed a series of E1/E1L–GFP fusion constructs (Fig. 3b) and tracked their subcellular distribution by transient expression in *N. benthamiana* leaves. Confocal imaging showed that E1–GFP was distributed primarily in the nucleus. E1La–GFP and E1Lb–GFP were distributed in both the nucleus and the cytoplasm, exhibiting a distribution pattern similar to that of e1-as whose NLS has undergone mutation. When we replaced the N-terminal peptide of E1La/E1Lb with the N-terminal peptide of E1 (designated 'E1–E1La–GFP' and 'E1Ω–E1Lb–GFP'); both proteins were exclusively localised to the nucleus, indicating the N-terminal peptide influences nuclear-localisation. We therefore shifted the N-terminal peptide of E1La/E1Lb to E1 (designated 'E1La/bΩ–E1–GFP') and the fusion localised to both the nucleus and cytoplasm, like E1La–GFP and E1Lb–GFP (Fig. 3c, Supplementary Fig. 6a–c).

To investigate which substitution/s caused the difference in nuclear localisation, we fused eGFP to four *E1* genes with different amino-acid substitutions in their N-terminal sequences. Confocal microscopy showed that only T12K led E1–GFP to localise in both the nucleus and cytoplasm, while the other substitutions did not affect nuclear localisation (Supplementary Fig. 7a–d). These results suggested that mutations at position 12 in the N-terminus of E1La and E1Lb affect nuclear localisation and reduce their regulatory activity, while E1 localises exclusively to the nucleus to strongly suppress *FT2a* and *FT5a*. Consistent with this, transient reporter-expression assay showed that E1 imparted stronger inhibition on the *FT2a* promoter activity than E1La and E1Lb (Supplementary Fig. 7e), and E1^T12K more weakly inhibited *FT2a* promoter activity compared to E1 and isoforms with substitutions in the N-terminus (Supplementary Fig. 7f). These results therefore partly explain the difference in flowering time governed by *E1* and two *E1-LIKE* genes.

In conclusion, the E1La and E1Lb amino-acid sequences have changed compared with E1, thereby affecting their nuclear-localisation. E1 maintains full protein activity while E1La and E1Lb have weakened activity. The partially compromised ability to inhibit the *FT2a* gene by mutation of *E1La* and *E1Lb* suggested that the *E1* family has undergone subfunctionalisation.

### Transcriptional regulation amongst *E1* family members is complex

*E1* and its homologues display a genetic-compensation response[34]. Here, to block the interference caused by genetic compensation, we conducted droplet digital PCR (ddPCR) analysis on the three *E1* genes using Wm82 (*e1-as*) and its isogenic line Wm82-*E1*. *E1* was expressed more strongly than *E1La* and *E1Lb* (Fig. 3d) in Wm82-*E1*. Copy numbers for *E1La* and *E1Lb* transcripts in Wm82 (*e1-as*) were greater than those

in Wm82-*E1* (Fig. 3e). *E1* copy numbers were higher in Wm82 (*e1-as*) than in Wm82-*E1* (Fig. 3f). These observations prompted us to investigate whether *E1* exerts transcriptional inhibition on *E1La*, *E1Lb* and even on itself. To address this, we conducted a transient reporter-expression assay. As an effector, E1–GFP could suppress the promoter activity of *E1*, *E1La* and *E1Lb* (Supplementary Fig. 8a). ChIP–qPCR suggested that E1–3xFLAG associates with the promoters of *E1* and its two homologues (Fig. 3g–i). These results indicate that *E1* possesses self-repression capability and could also suppress its two homologous genes.

Given that E1-family proteins might bind to the same promoter elements and are functionally conserved, it is conceivable that E1La and E1Lb might also possess similar inhibitory capabilities like E1. E1La and E1Lb could indeed suppress promoter activities of *E1*, *E1La* and *E1Lb*, and bind to the promoters of three *E1* genes using transient reporter-expression and ChIP–qPCR assays (Supplementary Fig. 8a–g). These experiments suggested that E1, E1La and E1Lb all bear self-repression capability and the ability to suppress their homologous genes. Whereas the results of qRT–PCR assays indicate that in the *E1La*[8] and *E1Lb* NILs (*E1Lb*-Wm82 and *E1Lb*-DN50), *E1La* and *E1Lb* do not have the same strong inhibitory capability towards their homologous genes as *E1* does (Supplementary Fig. 8h). The statistically significantly higher expression levels of *E1*, coupled with the weaker nuclear-localisation ability of E1La and E1Lb, contribute to a dominant inhibitory effect of *E1* on both *E1La* and *E1Lb*.

These results also suggest the presence of a rare regulatory module composed of the *E1* homologues: *E1* maintains a central role and directly suppresses its homologues *E1La* and *E1Lb*. With the addition of this module, the model of soybean photoperiodic flowering pathway under long-day photoperiods has expanded: expression of *E1*, *E1La* and *E1Lb* is promoted by *E3/E4* and *E2*. In the middle layer, *E1* suppresses the transcription of *E1La* and *E1Lb*. Subsequently, three *E1* homologues inhibit *FT* expression and flowering time in a hierarchical manner. When *E1* is impaired, two relatively weak *E1* like genes (*E1La* and *E1Lb*) are derepressed, resulting in earlier flowering. If all three *E1* homologues are impaired simultaneously, soybean will exhibit a remarkable early-flowering phenotype (Supplementary Fig. 9).

### The evolutionary trajectory of *Tof4b*

Analysis of polymorphisms in the promoter and coding regions of *Tof4b* identified four high-confidence haplotypes, designated H1–H4 (Fig. 4a, b). Haplotype H1 is the functional *Tof4b* allele and the most abundant haplotype, H2 (*tof4b-1*) is the *tof4b* allele characterised above with the 50-bp promoter deletion, and H3 (*tof4b-2*) is a known loss-of-function haplotype with a 1-bp deletion causing a frameshift resulting in a truncation to 73 amino acids in the 192-amino-acid for Tof4b protein in addition to the aforementioned 50-bp promoter deletion[26]. H4 has a base substitution at nucleotide position 159 in the coding region, causing a substitution from serine to arginine at residue 53. Alignment of E1 homologues revealed that the conserved amino acid at residue 53 is serine, indicating that H4 constitutes a mutant allele

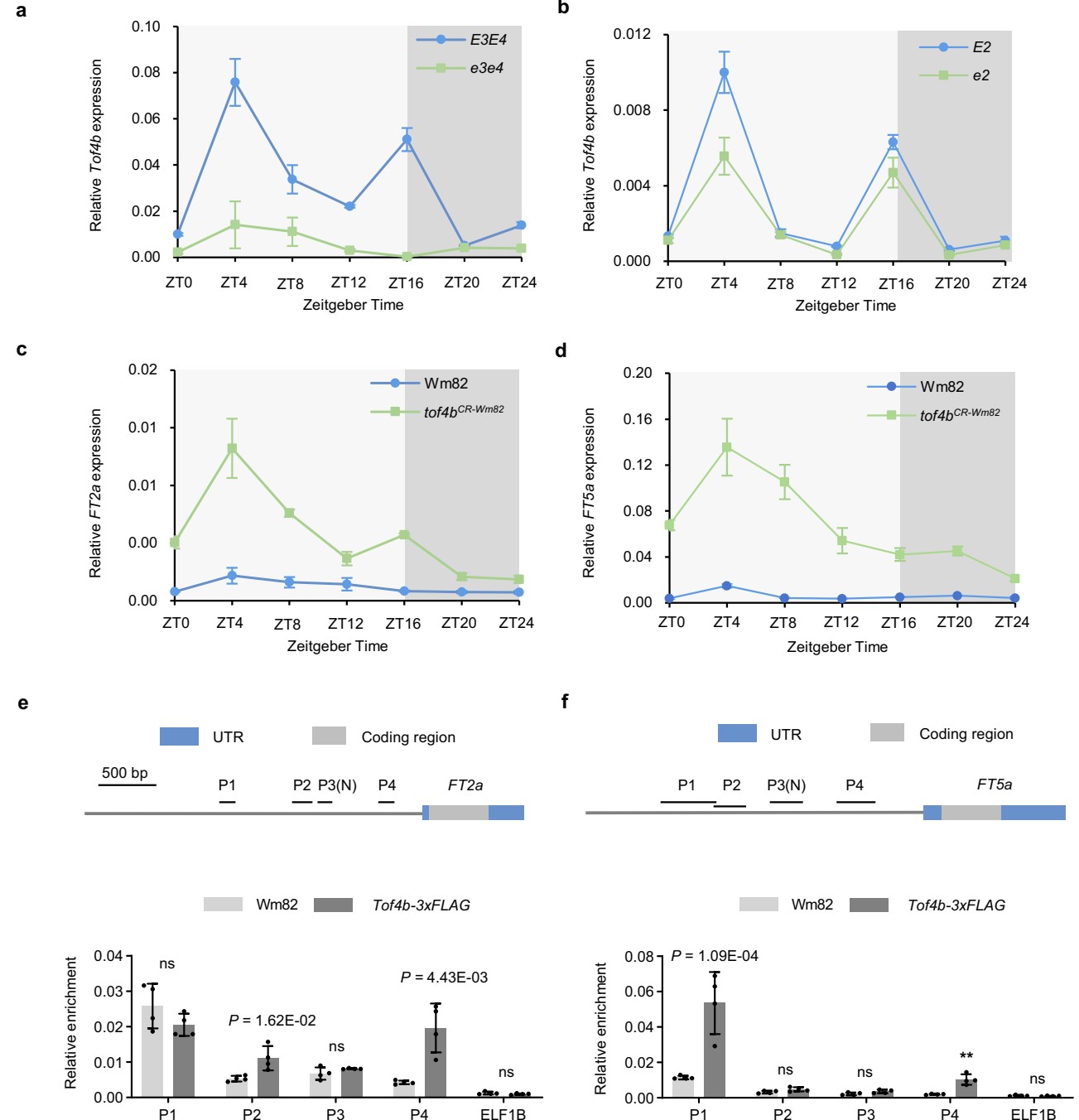

**Fig. 2 | *Tof4b* is up-regulated by *E3*/*E4* and *E2* to directly represses the expression of major florigens *FT2a* and *FT5a*.** Diurnal variation in *Tof4b* expression levels in NILs of *E3*/*E4* (**a**) and NILs of *E2* (**b**) under long-day photoperiod in a growth chamber. Data are the mean ± S.D. of n = 3 biological replicates and relative expression was normalised to *Tubulin*. Diurnal variation in *FT2a* (**c**) and *FT5a* (**d**) expression levels in Wm82 and Wm82-*tof4b*[CR-Wm82] under long-day photoperiod in a

growth chamber. Data are the mean ± S.D. of n = 3 biological replicates and expression was normalised to *Tubulin*. ChIP−qPCR assay of Tof4b-3xFLAG enrichment at the promoter region of *FT2a* (**e**) and *FT5a* (**f**). *ELF1b* was used as negative control. Data are the mean ± S.D. of n = 4 biological replicates. The two-sided Student's *t*-test was used to generate the *P* values. ns indicate no significant difference (*P* > 0.05). Source data are provided as a Source Data file.

(Supplementary Fig. 10a). A transient reporter-expression assay with *FT* promoters suggested that *Tof4b-H4* is a weak allele, consequently, it was designated *tof4b-3* (Supplementary Fig. 10b−d). Median-joining network analysis indicated that *tof4b-1* and *tof4b-3* might be derived from *Tof4b*, and that *tof4b-2* is derived from *tof4b-1* (Fig. 4b).

Alleles with the 50-bp promoter deletion (*tof4b-1* or *tof4b-2*) are present in cultivated soybeans (landraces and cultivars) widely across China, and in wild soybeans in northern China, but are absent from wild soybean accessions originating from the core domestication region of Huanghuai at middle latitudes (Fig. 4c). This implies that alleles with this mutation in cultivated soybean originated from an introgression from wild soybeans in northern China, rather than occurring later during domestication and improvement. A phylogenetic tree was constructed using re-sequencing data for the *Tof4b* region (2 Mb) from wild and cultivated soybeans in northern China. Cultivated accessions carrying the 50-bp promoter deletion clustered together with wild accessions carrying the deletion, and they collectively cluster with wild rather than with cultivated accessions

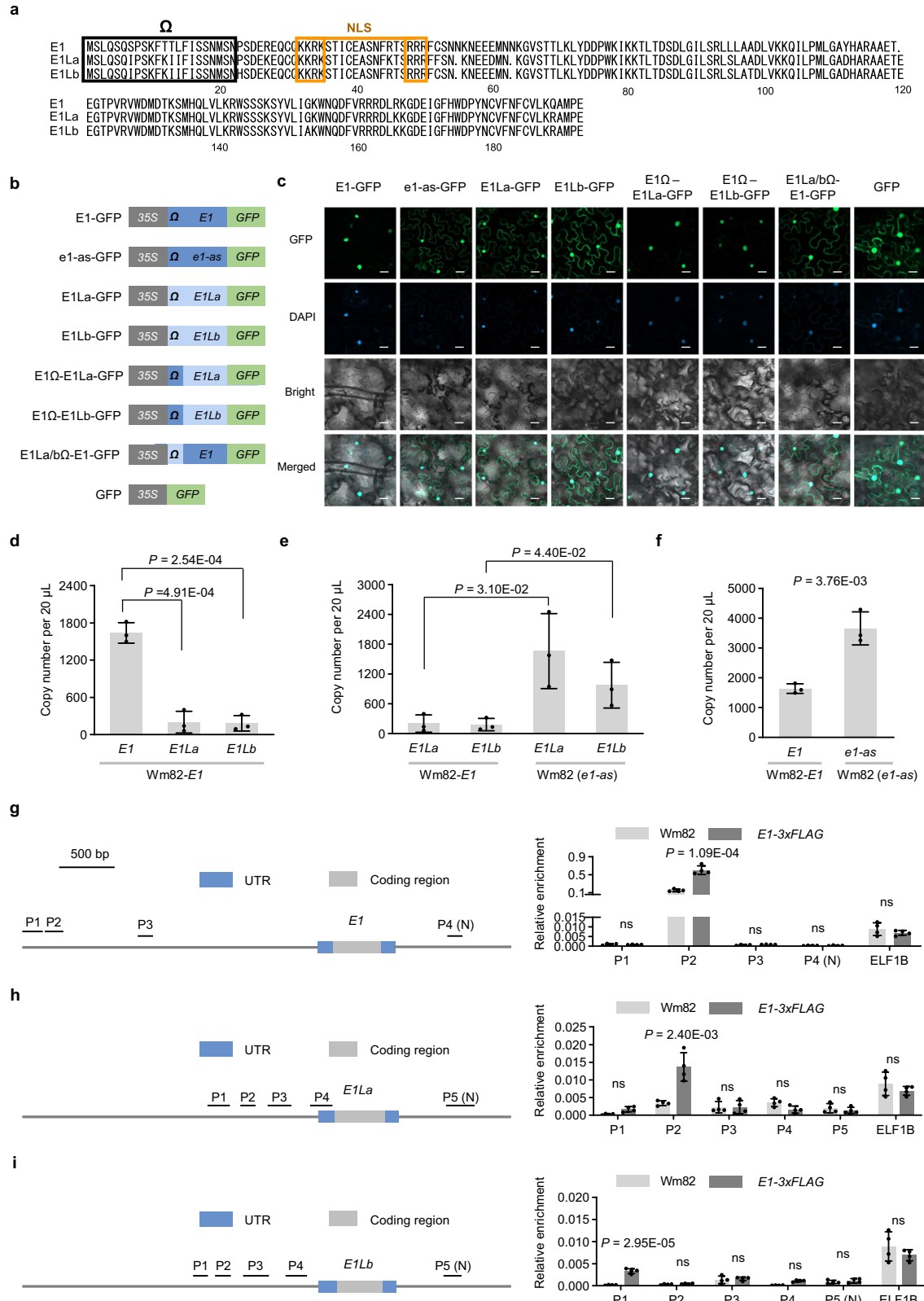

(Supplementary Fig. 11, Supplementary Data 2). These observations suggested that the 50-bp deletion mutation in cultivated accessions originated from an introgression from wild soybeans.

To further confirm and demarcate the segment carrying the promoter deletion of *Tof4b* introgressed into cultivated soybean, we calculated genetic diversity ($F_{ST}$ values) across chromosome 4 between cultivated soybean with the *tof4b-1/2* alleles and wild soybean

with *tof4b-1*, cultivated soybeans with *tof4b-1/2* and cultivated soybean with *Tof4b*, respectively. The $F_{ST}$ value curve of cultivated soybeans with *tof4b-1/2* and wild soybeans with the *tof4b-1* allele has a conspicuous trough between 16.1–26.3 Mb, suggesting that in this region, cultivated soybeans with *tof4b-1/2* and wild soybeans with *tof4b-1* have extremely low genetic differentiation. In the same region, differentiation between cultivated soybeans with *tof4b-1/2* and cultivated

**Fig. 3 | Subfunctionalisation of soybean E1 proteins. a** Alignment of full-length E1 orthologues from Wm82-E1. NLS represents nuclear-localisation signal. The numbers indicate every 20 residues. 'Ω' depicts the N-terminus peptide of E1 family proteins. **b** Schematic of the constructs used to assay subcellular localisation of soybean E1 proteins by confocal microscopy in transiently transgenic *N. benthamiana* leaves. GFP (Green Fluorescent Protein) was fused to C-termini and constructs were under the control of the *CaMV35S* (*Cauliflower Mosaic Virus 35S*) promoter. 'Ω' depicts the N-terminus peptide of E1 family proteins. **c** Confocal-microscopy analysis of subcellular localisation of soybean E1 proteins and their variants as summarised in (**b**). Scale bar = 25 µm. Three independent biological replicates were performed. **d** ddPCR comparison of *E1*, *E1La* and *E1Lb* expression in Wm82-*E1* at ZT 4 under long-day conditions at 20 DAE in a growth chamber. Data are the mean ± S.D. of n = 3 biological replicates. The two-sided Student's *t*-test was used to generate the *P* values. **e** ddPCR comparison of *E1La* and *E1Lb* expression in soybeans with *Tof4b* is much higher (Fig. 4d). A map of SNPs on chromosome 4 corroborates the pairwise $F_{ST}$ comparisons (Fig. 4e, Supplementary Data 3). This indicates that cultivated soybeans with the *Tof4b* promoter deletion originated from an introgression, during which the 10.2-Mb segment spanning from 16.1–26.3 Mb on chromosome 4 was introgressed from wild soybeans carrying the deletion. Moreover, we investigated nucleotide diversity (Pi) on chromosome 4 and observed that wild and cultivated soybeans carrying *tof4b1/2* exhibit significantly lower nucleotide diversity in this region compared to those carrying *Tof4b*, implying that *tof4b1/2* is a relatively recent mutation (Supplementary Fig. 12). In summary, we propose the evolutionary trajectory of the *Tof4b* gene was as follows: in high-latitude regions, wild soybeans have two mutations, a 50-bp promoter deletion and amino-acid substitution, corresponding to *tof4b-1* and *tof4b-3*, respectively. Of these, the *tof4b-1* allele was introgressed first into cultivated soybeans in these regions, wherein it subsequently evolved *tof4b-2* (Fig. 4b).

Wm82-*E1* and Wm82 (*e1-as*) backgrounds at ZT 4 under long-day conditions at 20 DAE in a growth chamber. Data are the mean ± S.D. of n = 3 biological replicates. The two-sided Student's *t*-test was used to generate the *P* values. **f** ddPCR comparison of *E1* expression in Wm82-*E1* and Wm82 (*e1-as*) backgrounds at ZT 4 under long-day conditions at 20 DAE in a growth chamber. Data are the mean ± S.D. of n = 3 biological replicates. The two-sided Student's *t*-test was used to generate the *P* values. ChIP–qPCR assay for enrichment of E1–3xFLAG at the promoter of *E1* (**g**), *E1La* (**h**) and *E1Lb* (**i**). ELF1b served as a negative control. Schematics of promoters and coding regions are shown on the left with the amplified regions depicted (P1–3). 'N' depicts negative control. Data are the mean ± S.D. of n = 4 biological replicates. The two-sided Student's *t*-test was used to generate the *P* values. ns indicate no significant difference (*P* > 0.05). Source data are provided as a Source Data file.

soybeans carry the *tof4b-2* allele. Taken together, *tof4b* alleles were subjected to strong artificial selection because they greatly improve high-latitude adaptation of cultivated soybeans, thus driving the spread of cultivated soybean to these latitudes.

## The genetic basis of wild soybean adapting to high latitudes

To identify whether the *Tof4b* allele contributes to the adaptation of wild soybeans to high latitudes, we examined the association between *Tof4b* genotype and flowering time using a panel of 254 wild soybean accessions grown in Harbin. Similar to cultivated accessions, wild accessions with *tof4b-1* or *tof4b-3* alleles flowered statistically significantly earlier than accessions carrying *Tof4b* (Supplementary Fig. 13c, d).

We then investigated the geographic distribution of the major *Tof4b* alleles in a panel of 591 sequenced wild soybean accessions found in northern China, Russia, Japan and the Korean peninsula. As expected, all the sampled wild soybeans from Japan and mid- and low-latitude regions of China carry the functional *Tof4b* allele. In northern China, 55% of the sampled accessions carry weak *tof4b* alleles, of which 45% are *tof4b-1* and the remaining 10% is *tof4b-2*. Moreover, the proportion of *tof4b-1* allele carriers in wild soybeans in Russia reaches 66%. These analyses confirm that *tof4b* alleles have been subjected to natural selection and thus contribute to the adaptation of wild soybean to high latitudes (Fig. 5b).

*Tof4*, *Tof5* and *E3* also contribute to the adaptation of wild soybeans to high latitudes[8,25,27]. To gain further insight into the genetic basis of wild soybean adapting to these latitudes, we simultaneously examined the allelic distributions of a series of known high-latitude-adaption genes of cultivated and wild soybean. The geographic distribution of *Tof4*, *Tof5* and *E3* is consistent with the present study and validates the proposed roles of *tof4*, *e3* mutants and the *Tof5^H2* allele in expansion toward higher latitudes for wild soybeans (Fig. 5c–e)[8,27].

Although most wild soybeans carry *E1* with a normal coding sequence[27], we still consider *E1* to play a major role in high-latitude adaption of wild soybean because of its strong effect on flowering time[20]. We, therefore, evaluated the *E1* promoter in the 591-accession panel and found the promoters could be roughly assigned based on the nucleotide polymorphisms (6 SNPs and 4 InDels) between two groups, Group I and Group II (Supplementary Fig. 14a). Using transient-reporter assays, we found that Group-I promoter activity is stronger than Group II (Supplementary Fig. 14b, c). We also examined the expression levels of *E1* in ten wild soybeans carrying the *E1* and *e1-wp* alleles. We found differences in the mean *E1* expression levels between accessions carrying the *E1* and those carrying *e1-wp* (Supplementary Fig. 15a, Supplementary Table 1). This suggests that the *E1* allele with a Group-II promoter is another weak *e1* allele in addition to *e1-as*, *e1-fs*, *e1-nl* and *e1-b3a*[21], which we have assigned *e1-wp* (*e1-weak promoter*).

We then explored the latitudinal distribution of different *E1* promoters. In high latitudes, the Group-II promoter is present at a frequency of about 50% in accessions from Russia and northern China, 20% in Korea and Japan, and is almost absent in the core domestication

## Mutations in *Tof4b* drove the spread of soybean to extremely high latitudes

Recent studies have identified and characterised a series of genes related to soybean adaption to high latitudes[3,5,19,23,24,35–37]. Our findings show that *Tof4b* may also play essential roles in the adaptation of cultivated soybean to high latitudes. To verify the effects of the introduction of *tof4b*, we examined its association with flowering time at Harbin (45° N) for 2 years. In a genetic background suited to high latitudes (*e1 e2*), cultivated soybean accessions carrying *tof4b-1* alleles flowered statistically significantly earlier than cultivated soybeans carrying the *tof4b* allele. Cultivated soybeans carrying *tof4b-2* showed distinctly early flowering out of the three alleles (Supplementary Fig. 13a, b). These phenotypes suggest that certain *tof4b* alleles, especially *tof4b-2*, can further improve adaptation to extremely high latitudes in an *e1 e2* genetic background, which is already adapted to high latitudes to a certain extent.

Next, we examined the global distribution of *Tof4b* alleles in cultivated soybeans (Fig. 5a, Supplementary Data 4). *tof4b* alleles are almost absent in countries and regions at low latitudes, including southeast Asia, south Asia, Africa, Australia and southern China. The only exception is Brazil, where the *tof4b-1* allele has a considerable degree of enrichment (allele frequency of 32%). A potential reason might be that the accessions in Brazil are all in the *E1* background, and *E1* could largely mask the effect of *e1lb*, thus soybeans in this region have no selection pressure on *E1Lb*. In contrast to the situation at low latitudes, the frequency of *tof4b* alleles increases with increasing latitude at middle and high latitudes. Nearly half of the surveyed Russian soybeans carry *tof4b* alleles, and *tof4b-1* accounted for 24% of varieties in Europe. A considerable percentage of *tof4b* alleles are also present in northern China, Canada, Europe and Kazakhstan. Furthermore, the least-functional allele *tof4b-2* is enriched in extremely high latitudes, such as Russia, Kazakhstan and Canada. Notably, 15% of Russian

**a**

| Haplotypes | Alleles | Promoter | | Coding sequence | |
|---|---|---|---|---|---|
| H1 | *Tof4b* | TGATCAATTCGTATAGAGATATATTTCTGTAGTAATTAATAAAAAACAA | | C | C |
| H2 | *tof4b-1* | - | | C | C |
| H3 | *tof4b-2* | - | | C | - |
| H4 | *tof4b-3* | TGATCAATTCGTATAGAGATATATTTCTGTAGTAATTAATAAAAAACAA | | A | C |
| Number of nucleotides (Wm82) | | -1538 ~ -1587 | | 159 | 219 |
| Number of amino acids (Wm82) | | | | 53 | 73 |
| Amino acid identified in Wm82 | | | | S | D |
| Amino acid change | | | | R | Frame shift |

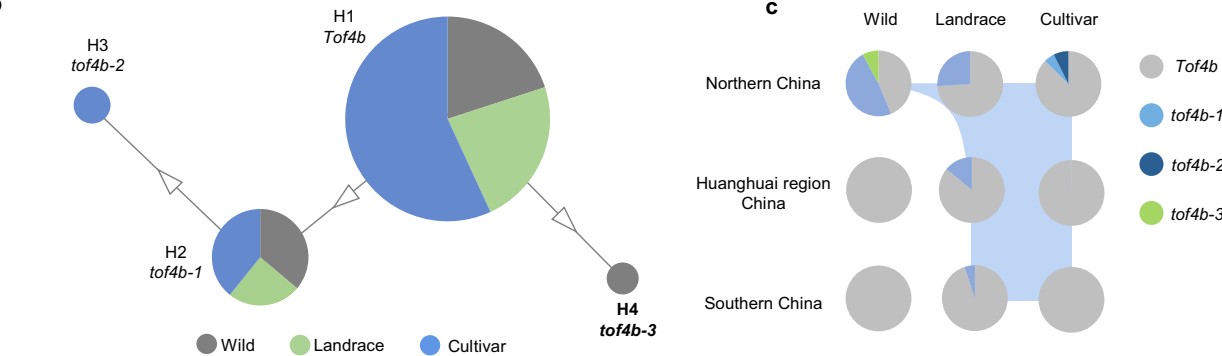

**b**

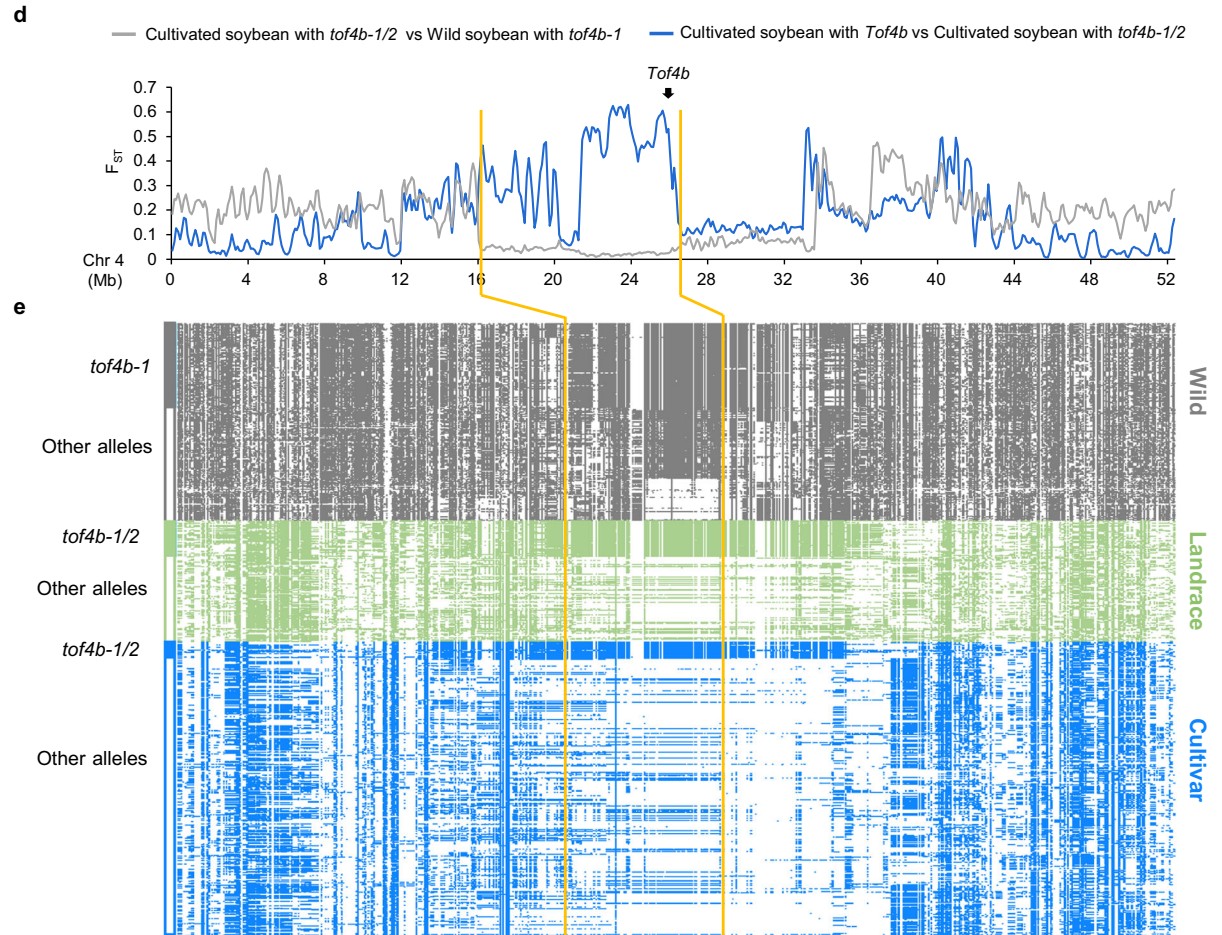

**Fig. 4 | The evolutionary trajectory of *Tof4b*. a** Summary of *Tof4b* haplotypes based on polymorphisms in the promoter and coding region. Dark blue represents a deletion in the promoter or coding sequence. **b** Haplotype origins of *Tof4b*. Grey represents wild soybean accessions, green represents landraces, blue represents improved cultivars. Triangles represent mutations that lead to weakened or loss of function. H1–H4 correspond to the haplotypes described in (**a**). **c** Allelic distribution of *Tof4b* in wild soybeans, landraces and cultivars in China according to region of origin. **d** Comparison of $F_{ST}$ values for chromosome 4 from cultivated soybean with *Tof4b* vs cultivated soybean with *tof4b-1/2*, and cultivated soybean with *tof4b-1/2* vs wild soybean with *tof4b-1*. The red arrow represents the location of *Tof4b*. **e** High-confidence mutations on chromosome 4 for wild soybeans, landraces and cultivars. Coloured cells represent SNPs or indels based on comparison with the Wm82 reference genome. The y-axis indicates six subgroups. The orange lines correspond to the introgression segment.

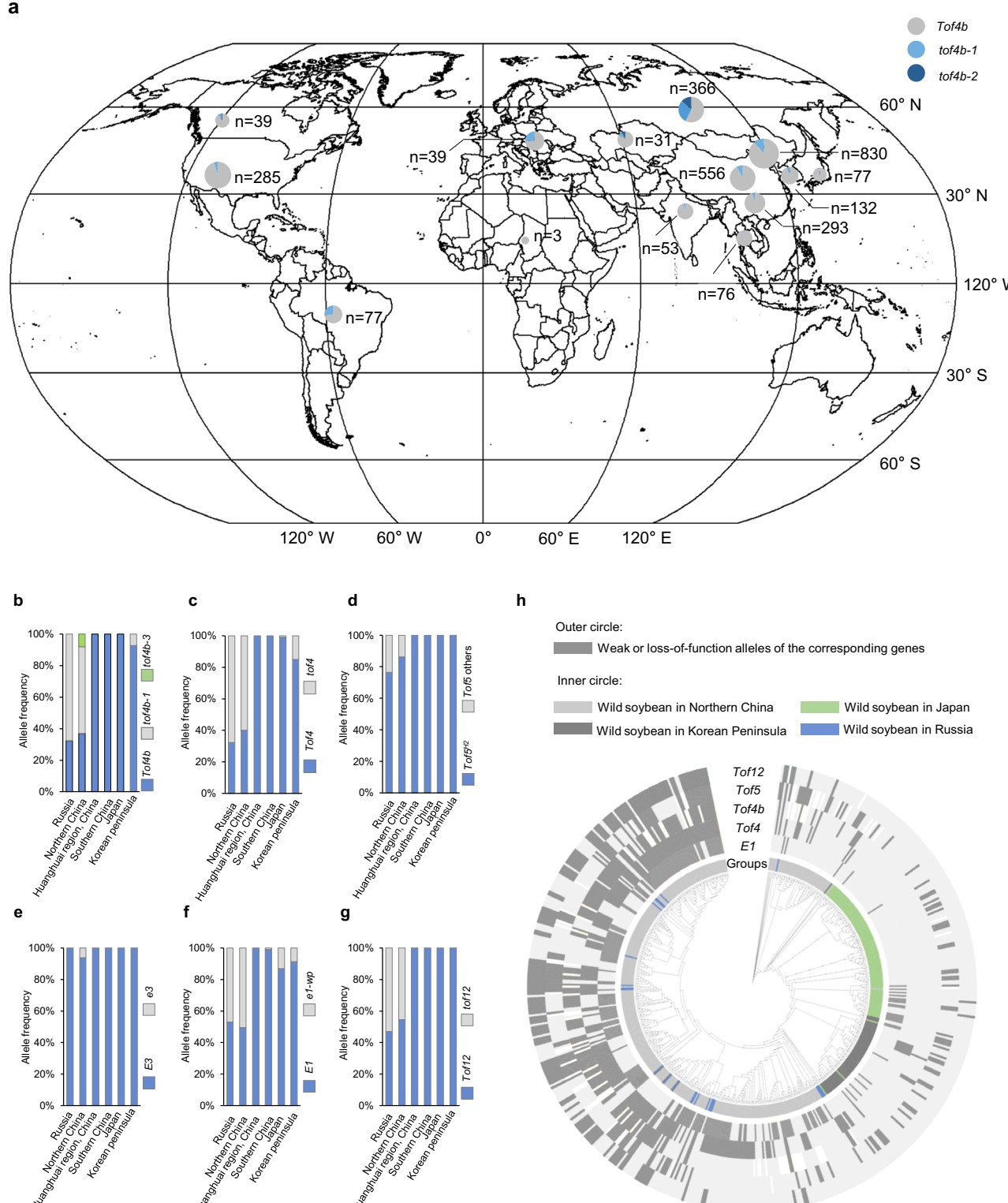

**Fig. 5 | Geographic distribution of *Tof4b* genetic diversity and the genetic basis of adaptation to high latitudes by wild soybean. a** Geographic distribution of *Tof4b* genetic diversity in cultivated soybean. Pie charts depict the relative abundance of each haplotype at each location. The map was drawn using ArcGIS v.10.3 software for desktop (https://desktop.arcgis.com/en/). Geographic distribution of genetic diversity of *Tof4b* (**b**), *Tof4* (**c**), *Tof5* (**d**), *E3* (**e**), *E1* (**f**) and *Tof12* (**g**) in wild soybeans. Numbers of accessions in each group are 34 (Russia), 333 (northern China), 39 (Huanghuai region, China), 61 (southern China), 65 (Japan), 23 (Korean peninsula). **h** Geographical distribution of genetic diversity of *E1*, *Tof4*, *Tof4b*, *Tof5* and *Tof12* in wild soybeans.

region of Huanghuai and southern China (Fig. 5f). These results suggest that alleles with mutations in the *E1* promoter (*e1-wp*) are also important for high-latitude adaption for wild soybean, and the *e1-wp* allele can be further introgressed into elite cultivars from wild soybean

for enhanced adaptability. We also found an abundant loss-of-function allele in the *Tof12* gene. This allele involves a C→T mutation at nucleotide position 43 that results in a premature stop codon, referred to as *tof12-2*[5]. The *tof12-2* allele is distinct from the *tof12-1* allele

(a stop-gain mutation at 1879 bp) fixed in cultivated soybeans during domestication[5], and only presents in wild soybeans found at high latitudes and is completely absent in cultivated soybeans. *tof12-2* carriers occupy a large percentage of the wild soybeans from northern China and Russia, suggesting that the loss-of-function of *Tof12* also contributed to the adaptation of wild soybean to high-latitude regions (Fig. 5g). Taken together, more than 92% of wild soybean accessions harbour at least one mutant allele in *E1*, *Tof4* (*E1La*), *Tof4b* (*E1Lb*), *Tof5* (*FUL*) and *Tof12* in the high-latitude regions of northern China and Russia. Mutations in the three *E1* genes themselves, their enhancer (*Tof12*) and downstream gene (*Tof5*), were selected and incorporated to condition wild soybean for high latitudes (Fig. 5h, Supplementary Data 5).

## Discussion

As an extremely photoperiod-sensitive plant, soybean's latitudinal expansion necessitates the loss of photoperiod sensitivity[38,39]. Reducing photoperiod sensitivity and reshaping flowering time have always been a major focus during soybean domestication, improvement and diversification. Previous studies have characterised a series of genes conferring adaptation of cultivated soybean to high latitudes[3]. However, the genetic basis for the expansion of wild soybean towards high-latitude regions remained poorly understood. Here, we have identified the *Tof4b* gene and demonstrated that its impairment facilitates adaptation to high latitudes for both wild and cultivated soybean. The early-flowering *tof4b* alleles, especially *tof4b-2*, have potential in molecular breeding to improve adaptation in extremely high latitude regions. In addition, the coding region of *Tof4b/E1Lb*, together with its homologues *E1* and *E1La*, have undergone subfunctionalisation during soybean genome duplication and evolution. Additionally, three *E1* genes all possess self-repression capability and can suppress their two homologous genes. These features form a rare regulatory module and participate in the soybean photoperiodic flowering pathway. Mutations in this pathway facilitated the expansion of wild soybean to high latitudes.

Cultivated soybean was domesticated from its wild progenitor *G. soja* and most genes/alleles were sourced from wild soybean in the domestication region or appeared during subsequent improvement and diversification[40]. While genetic bottlenecks during domestication lost approximately half of the genetic diversity[41], this inevitably led to missing many potential allelic variants in breeding that control key agronomic traits, including flowering time. Previous work found *Tof5^{H2}* and *Tof4^{H1}* facilitate high-latitude adaptation for wild soybean, but these alleles have rarely been applied to cultivated soybean[8,25]. Here, we found a weak *E1* allele (*e1-wp*), and a loss-of-function allele of *Tof12* (*tof12-2*) from wild soybean, which have not been used in breeding programmes, based on our analyses. We also identified an allele named *tof4b-1* that confers adaptation to wild soybean at high latitude and has been introgressed to cultivated soybean and thereby drove the spread of cultivated varieties. This phenomenon is not uncommon, as evidenced by instances of gene introgression from corresponding wild species into the genomes of cultivated varieties of other plants, such as maize, sunflower and rice[42–44]. Such genetic introgression serves to enhance the genetic diversity of cultivated varieties, equipping them with the ability to adapt to diverse environments. Further exploration is needed to identify other introgressed genes in the soybean genome and understand their biological functions.

Whole-genome duplications are widespread in plants and tend to generate gene duplications. Evolutionary processes suggest three alternative fates for the duplicate homologues: non-functionalisation, neofunctionalisation and subfunctionalisation[45,46]. Duplicated genes can be preserved by neofunctionalisation and subfunctionalisation, thus generating biological novelty and diversity[45–47]. In soybean, two whole-genome duplications occurred at ~59 and 13 million years ago (Mya), resulting in nearly 75% of the genes being present in multiple copies[48]. One such example is that *E1* has two homologues, namely *E1La* and *E1Lb*. The divergence time between *E1* and *E1La/E1Lb* can be traced back to 10.6 and 8.8 Mya, respectively[22]. However, there is scant understanding of the relationship and evolutionary fate of E1 homologues. In plants with CRISPR–Cas9-mediated *E1* mutations, *E1La* and *E1Lb* were up-regulated, indicating a common genetic-compensation response[34]. Such compensation phenomena are common in knockout mutations and are caused by H3 lysine 4 trimethylation (H3K4me3) triggered by mutant mRNA degradation[49], thus not reflecting the bona fide regulatory relationship between homologous genes. In this study, we quantified the expression of *E1*, *E1La* and *E1Lb* in Wm82 (*e1-as*) and Wm82-*E1* NIL lines instead of CRISPR–Cas9-mediated *E1* mutant alleles to avoid the interference from genetic compensation and performed transient assays and ChIP–qPCR to confirm the direct inhibitory effect of three *E1* genes on themselves, rather than from genetic compensation. Such self-repression has been reported for *REDUCED COMPLEXITY* (*RCO*)[50] and some *MYB* transcription factors[51]. A possible outcome of the self-repression of *E1* genes is to balance the total activity of the E1 homologues to an ideal level to finetune flowering and adaptation.

Together with the subcellular-localisation analysis, we uncovered a rare evolutionary fate for *E1* homologues. Firstly, the three *E1* genes all possess self-repression capability and can suppress their two homologous genes. While this may appear consistent with subfunctionalisation, defining it as such would be inappropriate if *E1* possessed self-suppression capability before genomic duplication. Determining whether this suppression capability emerged when *E1* first appeared in soybeans or after *E1* was expanded to three copies is for future studies. Secondly, the *E1* homologues have undergone subfunctionalisation by altering their subcellular localisation. The driving force of the evolution of the *E1* homologues might be avoiding extreme late flowering caused by the strong flowering suppressor, E1. Based on this model, *E1La* and *E1Lb* will be released from repression when *E1* function is impaired. Many cultivars adapted to high latitudes possess loss-of-function *E1* mutations. Moreover, in addition to mutations in *E1*, further variation in *E1La* or *E1Lb* will help enhance soybean's capacity to adapt to even higher latitudes. (Supplementary Fig. 16a, b).

The past few decades have seen substantial progress in understanding high-latitude adaptation of cultivated soybeans, while attention has seldom been paid to this for their wild progenitors. *Tof4-1*, *Tof5^{H2}* and loss-of-function alleles of *E3* all contribute to wild soybean high-latitude adaptation[8,25,27]. These genes alone, however, cannot fully explain high-latitude adaptation. Here, we identified *tof4b-1/3*, two contributors to high-latitude adaptation and further assessed the significance of several genes relating to cultivated-soybean adaptation, including *E1*, *E3*, *Tof4*, *Tof5* and *Tof12*. Among them, *Tof4* is located close to *Tof4b* on chromosome 4. We found that besides carriers of double-gene mutations, carriers of *Tof4/tof4b-1/3* are significantly fewer than those carrying *tof4/Tof4b*, hinting at the possibility that mutations in *tof4b* occurred later than mutations in *tof4*. Furthermore, the fact that 71.9 % of *tof4b-1/3* mutations occur together with *tof4-2*, rather than the 50% expected if the two genes were completely unlinked, indicates a degree of linkage between *tof4-2* and *tof4b-1/3*. Although *E1* negatively regulates *E1La* and *E1lb*, there is no obvious enrichment in mutations between *e1-wp* and *e1la* or *e1lb* or both (48.1% *e1la* carries are under *e1-wp* background, and for *e1la* is 46.8%) (Supplementary Fig. 17). Besides, it is intriguing to note that the expression level of *E1* in wild soybeans carrying *e1-wp* is indeed lower. Conversely, the expression level of *E1Lb* is elevated, while no statistically significant difference was seen for *E1La* expression (Supplementary Fig. 15b, c, Supplementary Table 1). This could possibly be attributed to the complexity of the genetic background.

In addition, we found that wild and cultivated soybeans generally selected different early-flowering alleles at *E1* and *Tof12*. Cultivated

soybeans enriched *tof12-1* alleles and several *e1* alleles with mutations in the protein-coding region, meanwhile wild soybean selected the *tof12-2* allele, which has a frameshift at 45 bp and the *e1-wp* allele with a weaker promoter. These two alleles from wild soybean have been lost during genetic bottlenecks associated with domestication and therefore have great potential for the molecular breeding of elite cultivars adapted to high-latitude regions.

In conclusion, we found that the *E1* gene family, the legume-specific transcription factor and core regulator of flowering in soybeans, has undergone subfunctionalisation and capable of self-suppression and mutual inhibition, shaping the regulatory ability of the E1 family in flowering. These findings provide new insights into the fate of duplicated genes and the control of flowering time in soybeans. Additionally, we discovered that mutations in the *E1* gene family, as well as in the genes *Tof5* and *Tof12*, enhance the adaptability of wild soybeans in high-latitude regions. These early-flowering alleles will hopefully contribute to advancements in soybean breeding for high-latitude regions.

## Methods

### Plant materials and growth conditions

The soybean diversity panel used in this study was grown from May to October under natural long-day conditions in 2020 and 2021 in Harbin, China (45° 750′ N, 126° 630′ E). Each row was 2 m long, with 60 cm spacing of between rows. 20 plants were sown in each row. Flowering time was recorded in 2020 and 2021. NILs of *Tof4b* were selected from $F_8$ progeny of this same cross using molecular markers for *Tof4*.

The NILs, CRISPR–Cas9 knockout mutants and overexpression lines were grown under long-day conditions (16 h light/8 h dark at 25 °C) in growth cabinets (Conviron Adaptis A1000) with a light intensity of 500 µmol photons $m^{-2} s^{-1}$.

### Resequencing, mapping, variation calling and haplotype-origin analysis

We used several published resequencing data or VCF files[5,8,18,27]. For each of the new accessions in this work, at least 5 µg of DNA was used to construct a sequencing library with an Illumina TruSeq DNA Sample Prep Kit, according to the manufacturer's instructions. Paired-end sequencing of each library was performed on an Illumina HiSeq X Ten system. Paired-end resequencing reads of the new accessions in this study were mapped to the reference genome (Gmax_Wm82_a2_v1) with BWA software with default parameters[52]. Filtration of duplicates of the sequencing reads, SNP and InDel calling was performed by VariantFiltration tools in GATK (4.1.1.0)[53]. The new VCF file in this study and published VCF files are merged by vcftools (0.1.16)[54]. Annotation was achieved by ANNOVAR (−0400)[55]. High-confidence mutations (MAF > 0.05, max missing <0.01) on chromosome 4 were called by vcftools to estimate the introgress interval of *Tof4b*. Mutations called by vcftools on the ~4-kb region upstream of *E1* were used for grouping the promoter of *E1* in wild soybeans from high latitudes. Haplotype-origin analysis was performed by Network 10.2 with median-joining method (https://www.fluxus-engineering.com/sharenet.htm).

### Construction of phylogenetic trees

To conduct the phylogenetic analysis, SNPs of selected accessions (wild and cultivated soybeans from high latitudes) were filtered with MAF = 0.05. These SNPs were used to construct an approximately-maximum-likelihood tree with FastTree (2.1) software[56] and were visualised with iTOL (https://itol.embl.de).

### DNA isolation and map-based cloning

Genomic DNA was extracted from fresh trifoliate leaves at 20 DAE with NuClean Plant Genomic DNA Kit (CWBIO) in accordance with the manufacturer's instructions. The primer sequences used to amplify the markers for mapping are listed in Supplementary Data 1. Markers were developed in the regions of *Tof4b* based on the resequencing data of the two parents, Wm82 and DN50. Recombinants were identified in the *Tof4b* fine-mapping population using seven markers. The flowering time of the progeny of these recombinants was evaluated to delimit the genomic interval containing *Tof4b*.

### Soybean transformation

*tof4b* mutants was created by CRISPR–Cas9 gene editing. Small-guide RNA sequences were cloned into pYLCRISPR/Cas9-DB vector to generate *Tof4b* CRISPR–Cas9 construct, which was then transformed into *Agrobacterium tumefaciens* EHA101 and used to transformed Williams 82 and DN50 plants using the cotyledonary-node method[37]. Material was selected on 8 mg/L Basta. To identify edited transformants, genomic DNA was first extracted from the leaves and specific primers were used to amplify then sequence the target sites to genotype the mutants. Primers used to construct the editing vector and identify the mutants are listed in Supplementary Data 1.

The CDS of *Tof4b* were amplified from Williams 82 and ligated into modified pTF101-3xFLAG vector under the control of the cauliflower mosaic virus *CaMV35S* promoter. The recombinant constructs were introduced into Williams 82 using the cotyledonary-node method as followed: First, isolate and sterilise soybean embryos. Second, infect them with Agrobacterium carrying target vector. Third, grow them on selective media to remove non-transformed cells. Fourth, cultivate surviving cells to develop shoots and roots. Fifth, transfer plantlets to soil and grow until mature. Transfer plantlets was selected on 8 mg/L Basta. To validate transgenic *Tof4b* overexpression plants, total protein from Wm82 and transgenic $35S_{pro}:Tof4b–3xFLAG$ lines were extracted for immunoblot analysis in extraction buffer (50 mM Tris-HCl pH 7.5, 150 mM NaCl, 5 mM EDTA, 0.1% v/v Triton X-100 and protease inhibitor cocktail). The anti-FLAG antibody (Sigma, M8823) was used in Immunoblot analysis was to detect transgene expression in the transgenic lines.

### RNA isolation and RT–qPCR

Trifoliate leaves were collected at 20 DAE at ZT 4 under long-day (16 h light/8 h dark) conditions. Each sample has three independent replicates. Total RNA was extracted using an Ultrapure RNA kit (CWBIO) and 0.5 µg was reverse transcribed using a Super Script First-strand cDNA Synthesis System (Takara, Dalian, China), all according to the manufacturers' instructions, respectively. Quantitative PCR (qPCR) was performed using SYBR Green Real-Time PCR Master Mix (Roche), according to the manufacturer's instructions. Three independent RNA samples were prepared for biological replicates. *Tubulin* (*Glyma.05G157300*) was used as the reference gene to normalise relative expression using the $2^{-\Delta\Delta CT}$ method[57]. All qPCR primers are listed in Supplementary Data 1.

### Droplet digital PCR

Droplet digital PCR (ddPCR) was used to analyse the absolute expression levels of *E1*, *E1La* and *E1Lb* in leaves of Wm82 and Wm82-*E1* under long-day conditions. The ddPCR was conducted in a 20 mL reaction mixture containing 10 mM of each target-gene primer and probe, 5 mL of ddPCR Supermix (dUTP), 10 ng cDNA and distilled water. The generation of nanoliter-sized droplets and PCR amplification were carried out using a MicroDrop-100A instrument (Forevergen, Guangzhou, China), following the manufacturer's protocol. The generation of nanoliter-sized droplets and PCR were performed in a MicroDrop-100A instrument (Forevergen, Guangzhou, China) following manufacturer's instruction. Each sample was obtained from three individuals and the data were analysed with three technical replicates.

### ChIP–qPCR

Leaf samples were collected at 20 DAE at ZT 4 under long-day conditions from Wm82 and $35S_{pro}:E1–3xFLAG$ transgenic lines. Samples

were fixed for 20 min in 1% v/v formaldehyde under vacuum using ice to cool samples down. The immunoprecipitation of soluble chromatin was done using anti-FLAG M2 magnetic beads (Sigma, M8823). The coimmunoprecipitated DNA was recovered and used for qPCR in triplicate. Relative fold enrichment was quantified by normalising the amount of a target DNA fragment against that of a genomic fragment of a reference gene, *ELONGATION FACTOR 1B* (*ELF1B*, *Glyma.02G276600*). Enriched of the *ELF1b* fragment was used as a negative control. Primers used for amplification are listed in Supplementary Data 1.

### Transient dual-reporter assays to assess promoter activity

The -3 kb *E1* promoter regions were amplified from the wild soybean accessions GDW061 and GDW099 (referred to Group I), ZKW0175 and ZKW0126 (referred to Group II). The four fragments were introduced into the pGreen0800-LUC/REN vector to generate the *35Spro-REN-pE1-LUC* reporter. -3-kb promoter sequences for *E1Lb* from Wm82 (P1 in Fig. 1d), ZKW0146 (P2 in Fig. 1d), GDW097 (P3 in Fig. 1d), DN50 (P4 in Fig. 1d) were also introduced into pGreen0800-LUC/REN to generate the *35Spro-REN-pE1Lb-LUC* reporters. Correctly assembled and sequenced vectors were introduced into *A. tumefaciens* GV3101 to grow overnight to $OD_{600\,nm} = 0.4–0.6$. Cell suspensions from effector and reporter constructs were added in equal amounts to infiltrate into fresh *N. benthamiana* leaves. At least three leaves from different *N. benthamiana* plants were infiltrated. LUC and REN activities were measured using a Luciferase 1000 Assay System (cat. no. E4550; Promega) and a Renilla Luciferase Assay System (cat. no. E2820; Promega), respectively. The final activity was expressed as the ratio between LUC and REN activities.

### Transient dual-reporter assays to assess transcriptional-regulatory activity

The -3 kb promoter sequences for *E1La* and *E1Lb*, *FT2a* and *FT5a* were amplified from Wm82. The four fragments were introduced into pGreen0800-LUC/REN vector to generate the *35Spro-REN-promoter-LUC* reporter[58]. *E1*, and different alleles of *Tof4b* (*Tof4b–H1*, *Tof4b–H3* and *Tof4b–H4*) were introduced into the pTF101–3Flag vector to generate the constructs *35S_pro:Tof4b–H1–3xFLAG*, *35S_pro:Tof4b–H3–3xFLAG* and *35S_pro:Tof4b–H4–3xFLAG* and were used as the effectors in the *N. benthamiana* transient-expression system[59]. Subsequent processing and testing are described above (Transient dual-reporter assays to assess promoter activity).

### Subcellular localisation and confocal microscopy

Coding sequences of all *E1*-family members without respective stop codons were fused with *GFP* in the pTF101–GFP vector driven by the cauliflower mosaic virus *CaMV35S* promoter[59]. The plasmid was transformed into *A. tumefaciens* GV3101, which was then infiltrated into *N. benthamiana* leaves for transient expression. The infiltrated plants were grown at 26 °C under long-day conditions (16 h light/8 h dark) for another 2 d before imaging. A Zeiss LSM 800 confocal laser-scanning microscope (Zeiss, Germany) was used to observe green fluorescent protein (GFP) fluorescence. *CaMV35S:GFP* was infiltrated as a localisation control.

DAPI was used to stain nuclei in 1x PBS at a working concentration of 100 ng/mL. *N. benthamiana* leaf samples were placed on a microscope slide to which a few drops of DAPI staining solution were and allowed to incubate for 10 min prior to imaging. Samples were observed under a confocal laser-scanning microscope with an excitation wavelength of 405 nm and emission wavelength of 460–500 nm.

### Cell-fractionation assays

Nucleus–cytosolic protein-fraction assays were performed as followed. Briefly, 200 mg *N. benthamiana* leaf samples were infiltrated with various *E1/E1La/E1Lb* expression plasmids for 48 h then frozen and ground in liquid nitrogen. Powdered leaf tissue was mixed with 500 µL of fraction lysis buffer (20 mM Tris-HCl pH 7.5, 25% v/v glycerol, 2 mM EDTA, 2.5 mM $MgCl_2$, 20 mM KCl, 250 mM sucrose, with the addition of 5 mM DTT with 1×protease inhibitor cocktail added fresh). The mixture was vortexed and filtered through a double layer of Miracloth (Millipore), and 40 µL sample was set aside as total protein. The flowthrough was centrifuged at 1500 *g* for 10 min at 4 °C, after which the supernatant was centrifuged at 10,000 *g* for 10 min at 4 °C, and the supernatant was collected as the cytosolic fraction. The pellet from the first centrifugation was washed five times with NRBT buffer (20 mM Tris-HCl pH 7.5, 25% v/v glycerol, 2.5 mM $MgCl_2$, 0.2% v/v TritonX-100) until it turned white in colour. The pellet was resuspended in 500 µL NRB2 buffer (20 mM Tris-HCl pH 7.5, 10 mM $MgCl_2$, 0.25 M sucrose, 0.5% v/v TritonX-100 and 5 mM β-mercaptoethanol supplemented with 1×protease inhibitor cocktail) and added on top of a layer of 500 µL NRB3 buffer (20 mM Tris-HCl pH 7.5, 10 mM $MgCl_2$, 1.7 M sucrose, 0.5% v/v Triton X-100 and 5 mM β-mercaptoethanol supplemented with 1×protease inhibitor cocktail added fresh). The suspension was centrifuged at 16,000 *g* for 45 min at 4 °C, and the pellet was collected as the nuclear fraction. Total protein, cytosolic proteins and nuclear fraction were resuspended in 2× SDS loading buffer and boiled for 5 min for immunoblot assays. E1–GFP fusion proteins were detected using anti-GFP antibodies (HT801-01; TransGen). PEPC was detected with anti-PEPC antibodies as the cytoplasmic marker (AS09458; Agrisera). Histone H3 was detected using anti-H3 antibodies as the nuclear marker (HL102-01; TransGen). All immunoblot bands were analysed for band greyscale values using ImageJ to estimate intensity as a measure of protein abundance. The calculation method for the proportion of proteins in nucleus and cytoplasm involves dividing the nuclear-protein intensity by the total-protein intensity and comparing it with the ratio of cytoplasmic-protein intensity to total-protein intensity, finally presenting it in the form of a percentage chart.

### Quantification and statistical analysis

All values were presented as the mean ± standard deviation (S.D.) and numbers (n) of samples or replicates are indicated in the respective figure legends. Data were analysed with GraphPad Prism 8 (ver. 8.0.1). Significance levels were calculated by the two-sided Student's *t*-tests or one-way ANOVA and presented by GraphPad Prism 8. To evaluate the phenotypes of the various accessions or lines, at least 10 individual plants of each accession were analysed. The n-values of statistical samples and the exact *P*-values are indicated in the figures

### Reporting summary

Further information on research design is available in the Nature Portfolio Reporting Summary linked to this article.

## Data availability

The sequencing data used in this study have been deposited in the NCBI SRA database under accession code SRP469369. The previously reported sequence data used in this study are available on the NCBI SRA database under accession codes SRP114890, SRP250886, SRP387668, SRP326758, SRP344122, and SRP387668. Source data are provided with this paper.

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

## Acknowledgements

This work was supported by the National Natural Science Foundation of China (32022062 and 32372112 to S. Lu, 31771815 to B.L., 32090064 to F.K., 32001503 to C.F., 32201865 to S. Li), the National Key Research and Development Program (2022YFD1201501 to F.K.), National Key Research and Development Program (2021YFF1001100 to S. Lu), and the Major Program of Guangdong Basic and Applied Research 2019B030302006 to F.K. and B.L., and the open-competition program of top-ten critical priorities of Agricultural Science and Technology Innovation for the 14th Five-Year Plan of Guangdong Province (2022SDZG05) to F.K. and B.L. Science and Technology Plan of Guangzhou, China (2023A04J1500 to C.F.).

## Author contributions

B. Liu, F. Kong and S. Lu designed and supervised the experiments and managed the projects. C. Fang, Z. Sun, S. Li, T. Su, L. Dong, L. Wang, H. Li, L. Kong, L Li, Z. Yang, F. He, Q. Cheng, F. Wang, M. He, X. Pei, X. Lin performed the experiments. S. Li, C. Fang, A. Zatybekov, and S. Lu performed the data analysis. C. Fang and S. Lu drafted the manuscript. B. Liu, F. Kong and S. Lu revised the manuscript.

## Competing interests

The authors declare no competing interests.
