## [Peer Review File · Nature Communications]

Subfunctionalisation and self-repression of duplicated E1 homologues finetunes soybean flowering and adaptationReviewers' Comments:

Reviewer #1:

Remarks to the Author:

Soybean, a short-day plant, is highly sensitive to photoperiod. Reduced photoperiod is essential for soybean to grow at higher latitudes. The group from Fanjiang Kong has conducted a series of comprehensive and systematic work demonstrating how cultivated soybean adapted to wide latitudinal regions. In this study, the authors further revealed how the subfunctionalization of E1 family genes and gene introgression drove soybean adaptation to higher latitudes. The authors identified E1Lb underlies a flowering time QTL Tof4b. A 50-bp indel in E1Lb promoter was verified as the causative variant. E1Lb is promoted by E2 and E3/E4 and directly suppresses the expression of FT2a and FT5a to delay soybean flowering under LD. The authors further demonstrated that E1 genes experienced two levels of subfunctionalisation either by differential subcellular localisation or direct transcriptional suppression of E1La and E1Lb by E1. The authors presented solid evidence that impairment of Tof4b facilitates adaptation to high latitudes for both wild and cultivated soybean. Especially, the widespread tof4b-1 allele in cultivated soybean originated from an introgression from wild soybean grown in high latitudes, clearly demonstrating the importance of gene flow in soybean evolution. In the end, the authors performed a comprehensive analysis for the genetic basis of wild soybean adaptation to high latitudes by combining results of six flowering genes. In summary, this manuscript is an elegant and remarkable work significantly advancing our understanding of the molecular and evolutionary mechanisms controlling soybean latitudinal adaptation. I only have a few minor questions as follows.

- Figure 1a, the middle graphical genotype did not align well with the left recombinant name and the right phenotypic analysis. I suggest the authors to adjust the display manners. All recombinants are homozygous for Wm82 at marker M01 and M10. I guess M01 and M10 are fixed background markers and thus provide limited information. To avoid confusion, I suggest to delete M01 and M10 in the figure. In Figure 1c, it would be better to use Wm82 and DN50 to name the NIL (Line 109).
- The statistical test for multiple comparison in Figure 1e and Supplementary Fig. 4 should be one-way ANOVA not student's t test. Please correct this.
- Williams 82 was sometimes designated 'W82', sometimes 'Wm82' throughout the manuscript and figures. The name should be consistent.
- Both SD and SE were used to indicate the data variation. I would suggest the authors to consistently use SD or SE throughout the manuscript.
- The second reporter in Supplementary Figure 1a should be pE1La (Wm82) not pE1La (DN50). The corresponding caption should be corrected.
- Line 166-167, should be "between e1-as E1La E1Lb and e1-as e1la E1Lb (4.6 days), e1-as E1la E1Lb and e1-as E1La e1lb (4 days)" according to Supplementary Figure 4.
- Both e1la and e1lb had a relatively small effect (~4 days) on soybean flowering time, while e1la e1lb flowered ~15 days earlier. Is there a genetic compensation between E1La and E1Lb?
- In Supplementary Figure 5, the reporter should be pFT2a not pPT2a.
- The Fst analysis showed that tof4b alleles were subjected to artificial or natural selection to help cultivated or wild soybean to adapt to high latitudes. Did the cultivated or wild soybean carrying tof4b alleles exhibit lower nucleotide diversity compared to the cultivated or wild soybean with Tof4b allele?
- In Figure 6, for the "early flowering model", the 'X' on the transcription of E1La and E1Lb should be deleted.
- Methods used for haplotype origin analysis were not presented in the "Materials and methods section".
- As the gene introgression from wild soybean driving cultivated soybean adaptation to high latitudes is an important discovery in this study, some further discussion on this topic would be helpful for readers.

Reviewer #2:

Remarks to the Author:

Fang et al. described that the *Tof4b* mutation is in the *E1Lb* gene. Also, the authors reported that the previously published *E1* transcript lacked a 5' end. They described the *e1-wp* mutation, which is the new allele of *E1*, and the *tof12-2* mutation (although details of this mutation are missing). In addition, *Tof4b* mutations are also enriched in wild soybeans adapted to higher latitudes. Although mapping the *Tof4b* mutation seemed to be a substantial amount of work, this paper appears to be an assortment of small incremental information (the mutation mapped to the known gene and the introduction of new alleles). Below are specific concerns that I hope to improve the current content.

Major comments/concerns:

1. Fig. 5 depicted that *tof4*, *tof4b*, *e1-wp*, and *tof12* mutations are all enriched in Russian and NC wild soybeans. Some of the information shown here was already published. What is missing here is an analysis of the relationships among those mutations to confer daylength insensitive phenotypes important for the flowering of soybeans in long days. All these mutations increase the expression of two FT genes in long days, so the mechanisms causing earlier flowering are the same. *Tof4* (*E1La*) and *Tof4b* (*E1Lb*) localized very close on the genome and functionally resembled each other. For example, are there any genetic links (ex., co-occurrence of specific mutations) between these two loci? As *E1* negatively regulates *E1La* and *E1Lb*, are there any enrichments of mutations between *E1-wp* and *E1La* or *E1Lb* or both? What is the relationship between *tof12* and *E1*, *E1La*, and *E1Lb* mutations? Are the *tof12* and *E1*, *E1La*, and *E1Lb* mutations alternative mutations that cause the same results, or do these mutations show additive effects (like the mutations among *E1*, *E1La*, and *E1Lb*)? Fig. 5h has the necessary information to answer these questions, but Fig. 5h by itself makes it difficult to assess these points. Since *Tof4b* is not the first gene reported to be important for higher latitude adaptation, it is crucial to understand and discuss the effects of other mutations that cause similar effects.
2. The authors described that *E1*, *E1La*, and *E1Lb* are subfunctionalized. I am not convinced of this statement (hence the title). All three genes are repressors of FT genes. Previous results showed that at least the long-day and short-day expression patterns of all three genes are very similar, and all of them are similarly regulated by *phyA* (*E3/E4*). The authors and others (Wang et al., 2022 *Frontiers in Plant Sci*) showed that *E1* represses *E1La* and *E1Lb*. Could *E1* bind to its own promoter to repress its expression (negative feedback)? If not, it could be a new connection, but if *E1* already has negative feedback, this mechanism might also be duplicated to *E1Ls* (which may indicate that *E1La* or *E1Lb* could negatively regulate *E1* and their expression as well). The authors need to check the possible connection of *E1* repressing *E1La* and *E1Lb* as a new pathway by studying the expression of these genes in corresponding lines. At least, based on the information present in this manuscript, there is not enough information to support the statement of subfunctionalization among *E1*, *E1La*, and *E1Lb*.
3. Mainly based on the results shown in Fig. 1, the authors proposed that the 50-bp deletion in the *E1Lb* promoter region is likely the cause of the *tof4b* phenotype. Could the authors discuss the presence of possible known cis-elements that are possibly important for the induction of *E1Lb* in the deleted 50-bp?
4. Also, regarding the Fig. 1c result, it would be more informative to understand the possible mechanism of the 50-bp deletion if the authors could analyze the *Tof4b* expression patterns in *Tof4b* and *tof4b* (*tof4b-2*) NIL lines in long days. Please analyze the daily *E1Lb* expression patterns in two NIL lines used for Fig. 1c in long days (like the ones shown in Fig. 2). This result would tell us whether the 50-bp deletion of the promoter reduces the expression of *Tof4b* (*E1Lb*) throughout the day (which means that an unknown activator for *E1Lb* may bind to the 50-bp region the whole day).
5. The *e1-wp* mutation is found in NC and Russian wild soybeans. If the *e1-wp* mutation is indeed the weak promoter mutation, the authors should show that *E1* level reduction in the *e1-wp* containing soybeans. In these lines, what is the expression of *E1La* and *E1Lb*? Are they elevated? Please analyze the expression of *E1*, *E1La*, and *E1Lb* in the *e1-wp* plants.

6. For discussing the effects of amino acid sequence differences in the N-terminal end on the nuclear localization, only showing GFP fluorescence is not that quantitative, as the exposure time difference or expression levels of GFP tagged protein could make the pictures look like nuclear enriched or not enriched as GFP signal by itself already strongly localized in the nucleus. To discuss the changes in nuclear and cytoplasmic distributions of different E1/E1L constructs, the authors should also perform fractionation (nuclear vs cytoplasm) and quantify using western blot analyses.

7. What is the *tof12-2* mutation? Fig. 5 shows that the *tof12-2* is enriched in soybeans from higher latitudes. The authors previously showed that *tof12-1* is enriched in soybeans from higher latitudes (Lu et al. Nature Gen 2020). *Tof12* mutations seem important for adapting to higher latitudes, but what is the relationship (allelic distributions) between these two mutations in soybeans from different areas? The authors mention that *tof1-2* has a frameshift mutation but without the data. Please add actual results to support these statements.

8. For Supplementary Figures 5b, e, and f, it is difficult to assess the strength of each effector construct by looking at the sude color of one transient assay image. The authors should show the averages of measurement values to discuss the strength of each E1/E1L-related effector on the reporter (like the results shown in Supplementary Figures 6c and d).

Minor mistake:

1. The X-axis level in Fig. 2C has a mistake (this should be the hours of the day). Please correct it.

Reviewer #3:

Remarks to the Author:

This is a worthwhile, interesting and wide-ranging study. Ranging from from QTL identification, gene editing, molecular mechanisms to evolutionary and adaptative aspects on three legume specific E1 flowering repressors in soybean with a major focus on E1Lb. These genes have been published on before, but this work lifts understanding of function and impact of the genes in soybean adaptation and cultivation.

Below are some issues to be commented on and a number of discrepancies to be corrected:

1. 68. adaptation is
2. 70. indicate that E2 is a soybean GIGANTEA
3. 78. sp FRUITFULL
4. 83. how *Tof4b*
5. 104. Explain that plants with A sequences ie from DN50, in region of interest, flower earlier than plants with B ie W82.
6. 108 . Explain which of the parents have the different mutations in E1Lb promoter ie . in text it indicates that the DN50 parent would have the deletion mutation.
7. 108. The text and figure 1 do not agree. On Fig 1, the W82 parent has the deletion mutation? This needs correcting.
8. 147 Add reference for statement about E1 and E1La binding the promoters of FT2a and FT5a.
9. 147. The text and figure 2 do not agree. On figure it is P1 and P4 fragments for both FT genes, in text it is P2 and P4 for FT2a. This needs correcting.
10. 181. The text and figure Supplementary 5b do not agree. Constructs 4+5 top right have more luminescence than 2+5 bottom right. This implies that the shorter E1 is a stronger repressor than the longer E1. This needs correcting.
11. 214. The text and figure 3e do not agree. It should read that ... expression levels of....are much higher thanThis needs correcting.
12. 215. The text and Supplementary Figure 5 are incorrectly referred to. Should be Figure 5g,h.
13. 244. Fig 5a needs correcting to Fig 4a.
14. 251. Phrase S53R differs ...seems out of place here.

15. 264 ..of wild andneed to insert missing adjective
16. 458. intensity of umol...not mmol
17. 511. qPCR not qRT-PCR
18. Fig 1a. – Indicate what the units are for flowering time in the graph. If Dae indicate what this stands for how flowering measured.
19. Fig 1e. Seems unusual that the promoter LUC fusions eg in Fig 1e and others, without any effectors co transfected, can give LUC expression in Benth leaves at the same level as the highly expressed 35S :REN? Please comment.
20. Fig 1b. As noted above, error in that DN50 has the deletion allele not W82.
21. Fig 1. Define **
22. Fig 3. Remove – different letters indicate..These letters are not shown.
23. Fig 4. Table in Fig 4a needs correcting and clarifying. Label the last columns. Also AA identified in W82 needs to swapped with AA# in W82.Define fs.
24. Figure 6 . Too small font. Define gray and blue in right hand panels.
25. Suppl fig 1. Co-transfection assay- but no effector is added- is that right? Again it is unusual that it is expressed strongly, especially the W82 promoter.
26. Suppl fig 1 The figure legend and the diagram to not agree. Diagram has different constructs that the legend? Diagram has pE1La promoter , while in legend it is proE1Lb (W82)? This needs correcting.
27. Suppl fig 3. Define DAE
28. Suppl fig 5 . Text and figure did not agree as above- Check in b that construct combinations correctly assigned.
29. Suppl fig 5b,e,f. Reporter should be pFT2a not pPT2a.
30. Suppl fig 5. As above comment on why strong expression of control FT2a promoter LUC without effectors in Benth?
31. Suppl fig 6b . v small font for Tof alleles.
32. Suppl Fig 8. Explain in fig legend and text why there are two Group 1 promoters and two group 2 promoters assayed in the figure. Are they the same or different from each other? What is HY107?. Needs explaining.
33. Suppl fig 9. Legend needs correcting. States: .. cultivated soybean in 851 wild soybeans. Which is it, cultivated or wild?

Reviewer #4:

Remarks to the Author:

Fang et al present a dense and comprehensive investigation of the soybean E1 gene family and target genes and alleles of those genes controlling adaptation to high latitudes. There are a multitude of new discoveries in this work that exemplify deep exploration of the genetic systems responsible for this important phenotype. Generally, I found the work scientifically suitable for publication in Nature Communications with one request for an expanded discussion for the possible authentic start site for the E1 gene (currently described without its own results heading and in Supplementary Figure 5).

Very Minor points:

Please refer to the gene IDs (Wm82.a2.v1 or other) at least once for all of the genes presented in the manuscript to avoid confusion of different naming schemes (for example, Tof4 [Glyma.04g156400]).

Line 68 "in" should be "is"

Line 95-96 is not complete sentence: "While Tof4 alone cannot fully explain the flowering differences caused by this interval."

Line 96-99 This sentence is also a little confusing and could be reworded.

Line 131 This section-what tissue was characterized? Missing from M&M.

Lines 155-173 Promoting Supp. Figure 4 to the main Results section might help with interpretation of the associated text.

Line 184-191 Complete sentence? Please reword.

Line 268 For Figure 4d, it was too small for me to see properly.

Line 413 Blue text?

Line 429 Please reword for clarity.

Line 433. Supp Figure 9 legend is unclear for last sentence. a is wild soybean and b is cultivated? The legend states "Data from cultivated soybeans in an ..."

Dear Reviewers,

Thank you all very much for your thoughtful and helpful comments on our manuscript that have assisted to strengthen the study.

According to your suggestions, we have made corrections and modifications according to the reviewers' comments and highlight the changes in the manuscript.

To Reviewer 1:

We are grateful for your advice. In response to your inquiries, we have provided with the following answers.

Question 1: Figure 1a, the middle graphical genotype did not align well with the left recombinant name and the right phenotypic analysis. I suggest the authors to adjust the display manners. All recombinants are homozygous for Wm82 at marker M01 and M10. I guess M01 and M10 are fixed background markers and thus provide limited information. To avoid confusion, I suggest to delete M01 and M10 in the figure. In Figure 1c, it would be better to use Wm82 and DN50 to name the NIL (Line 109).

Response: We appreciate your guidance. We have changed the Figure 1a and Figure 1c as your suggested and these changes have improved the clarity of the figures and should facilitate better understanding for our readers.

Question 2: The statistical test for multiple comparison in Figure 1e and Supplementary Fig. 4 should be one-way ANOVA not student's t test. Please correct this.

Response: Thank you for your professional comments. We have made the revisions as per your suggestions.

Question 3: Williams 82 was sometimes designated 'W82', sometimes 'Wm82' throughout the manuscript and figures. The name should be consistent.

Response: We apologize for these mistakes. We have corrected it as Wm82 in whole manuscript.

Question 4: Both SD and SE were used to indicate the data variation. I would suggest the authors to consistently use SD or SE throughout the manuscript.

Response: Thanks for your kind suggestion. In this manuscript, we only use SD to indicated the data variation throughout the manuscript. We have made the necessary adjustments accordingly.

Question 5: The second reporter in Supplementary Figure 1a should be pE1La (Wm82) not pE1La (DN50). The corresponding caption should be corrected.

Response: We apologize for the mistake and have corrected it.

Question 6: Line 166-167, should be “between e1-as E1La E1Lb and e1-as e1la E1Lb (4.6 days), e1-as E1la E1Lb and e1-as E1La e1lb (4 days)” according to Supplementary Figure 4.

Response: Thanks, we have changed those as suggested accordingly.

Question 7: Both e1la and e1lb had a relatively small effect (~4 days) on soybean flowering time, while e1la e1lb flowered ~15 days earlier. Is there a genetic compensation between E1La and E1Lb?

Response: We appreciate your suggestion to further validate the genetic compensation between *E1La* and *E1Lb*. We detected the expression of *E1La* in Wm82 and mutant of *e1lb*, and found that the expression level of *E1La* increased significantly. Similarly, the expression level of *E1Lb* in e1la mutant was significantly higher than that in the Wm82 (**Relevant Figure follow**). Consistent with your suggestions, these results indicated that there is genetic compensation between *E1La* and *E1Lb*.

On the other hand, we also found that E1La and E1Lb could bind each other's promoter to repress the expression (**Fig. 3g-i, Supplementary Fig. 8a-g**). Based on this model, mutations in one gene in either *E1La* or *E1Lb* not only amplify the effects of another gene by genetic compensation but also by partially relieving its inhibition. Both mechanisms contribute to the significant difference of flowering time of double

mutant, surpassing the simple cumulative of two single mutants.

Fig Expression of *E1La* and *E1Lb* in the *e1lb^{CR}* and *e1la^{CR}* mutant and Wild type (*Wm82*), respectively. Data are the mean \pm SD. $n = 3$. ** indicates statistically significant differences determined by Student's test ($P < 0.01$).

Question 8: In Supplementary Figure 5, the reporter should be pFT2a not pPT2a.

Response: We apologize for the mistake. We have corrected it accordingly.

Question 9: The *Fst* analysis showed that *tof4b* alleles were subjected to artificial or natural selection to help cultivated or wild soybean to adapt to high latitudes. Did the cultivated or wild soybean carrying *tof4b* alleles exhibit lower nucleotide diversity compared to the cultivated or wild soybean with *Tof4b* allele?

Response: We appreciate your comment on that. Upon further investigation into nucleotide diversity (π) within chromosome 4, we observed that wild and cultivated soybeans carrying *tof4b* exhibit significantly lower nucleotide diversity in this chromosomal region compared to those carrying *Tof4b* (**Supplementary Fig. 12**). We have incorporated this information into the results (Line 311-315).

Supplementary Fig. 12 | Pi within the wild soybeans, cultivated soybeans for chromosome 4 Pi within chromosome 4 of wild soybean with *Tof4b*, wild soybean with *tof4b-1*, cultivated soybean with *Tof4b*, and cultivated soybean with *tof4b-1&2*.

Question 10: In Figure 6, for the “early flowering model”, the ‘X’ on the transcription of E1La and E1Lb should be deleted.

Response: We deeply respect your rigorous attitudes. In our revised manuscript, we have refined the model and show it in Supplementary Fig. 9.

Supplementary Fig. 9 | Model of E1 family-members functions in the soybean photoperiodic flowering pathway. E1, E1La and E1Lb all possess the ability for self-repression and repression of homologous genes. Among these repression effects, the inhibition of E1 to *E1La* and *E1Lb* is predominant. Besides, the amino acid sequence for E1La and E1Lb have changed, affecting their nuclear localisation. E1 maintains the full protein activity while E1La and E1Lb maintain partial protein activity. Subsequently, three E1 homologues inhibit *FT* expression and flowering time in a hierarchical manner. The ellipse represents nucleus. This figure was created using BioRender (<https://biorender.com/>).

Question 11: Methods used for haplotype origin analysis were not presented in the “Materials and methods section”.

Response: We apologize for not including the methods for haplotype origin analysis in our manuscript. Haplotype origin analysis was performed by Network 10.2 with median joining method (<https://www.fluxus-engineering.com/sharenet.htm>). We have provided this information in the Materials and Methods of revised manuscript from line 539 to

line 541.

Question 12: As the gene introgression from wild soybean driving cultivated soybean adaptation to high latitudes is an important discovery in this study, some further discussion on this topic would be helpful for readers.

Response: Thanks for your nice advice. We have expanded some relevant discussion included at Line 441-447.

To Reviewer 2

Fang et al. described that the *Tof4b* mutation is in the *E1Lb* gene. Also, the authors reported that the previously published *E1* transcript lacked a 5' end. They described the *e1-wp* mutation, which is the new allele of *E1*, and the *tof12-2* mutation (although details of this mutation are missing). In addition, *Tof4b* mutations are also enriched in wild soybeans adapted to higher latitudes. Although mapping the *Tof4b* mutation seemed to be a substantial amount of work, this paper appears to be an assortment of small incremental information (the mutation mapped to the known gene and the introduction of new alleles). Below are specific concerns that I hope to improve the current content.

Response: Thanks for all your comments, which are valuable and very helpful. Your advice not only improve the quality of this paper, but also deepen our understanding. We have carefully incorporated all your comments and have made corrections accordingly.

Question 1: Fig. 5 depicted that *tof4*, *tof4b*, *e1-wp*, and *tof12* mutations are all enriched in Russian and NC wild soybeans. Some of the information shown here was already published. What is missing here is an analysis of the relationships among those mutations to confer daylength insensitive phenotypes important for the flowering of soybeans in long days. All these mutations increase the expression of two FT genes in long days, so the mechanisms causing earlier flowering are the same.

Tof4 (*E1La*) and *Tof4b* (*E1Lb*) localized very close on the genome and functionally resembled each other. For example, are there any genetic links (ex., co-occurrence of specific mutations) between these two loci? As *E1* negatively regulates *E1La* and *E1Lb*, are there any enrichments of mutations between *E1-wp* and *E1La* or *E1Lb* or both?

Response: In response to your first question regarding the genetic links between *Tof4* (*E1La*) and *Tof4b* (*E1Lb*), We found that besides carriers of double-gene mutation, carriers of *Tof4/tof4b-1&3* are significantly fewer than those carrying *tof4/Tof4b*, implying the possibility that mutations in *tof4b* occur later than mutations in *tof4*. Furthermore, the fact that 71.9 % of *tof4b-1&3* mutations occur together with *tof4-2*,

rather than the 50% expected if the two genes were completely unlinked, indicates a degree of linkage between *tof4-2* and *tof4b-1&3*. Although *E1* negatively regulates *E1La* and *E1Lb*, there is no obvious enrichment of mutations between *e1-wp* and *e1la* or *e1lb* or both (48.1% *e1la* carries are under *e1-wp* background, and for *e1la* is 46.8%)

In terms of the potential for mutation enrichment due to the negative regulation by *E1* on *E1La* (*Tof4*) and *E1Lb* (*Tof4b*), our analysis did not reveal any significant enrichment of mutations between the *e1-wp* and *e1la* or *e1lb* genes, or both. Specifically, 48.1% *e1la* and 46.8% *e1lb* carries are under *e1-wp* background, respectively (Supplementary Fig. 17).

Supplementary Fig. 17 | Venn diagram of relationship of mutants of *tof12-2*, *e1-wp*, *tof4-2* and *tof4b-1&3*.

What is the relationship between *tof12* and *E1*, *E1La*, and *E1Lb* mutations? Are the *tof12* and *E1*, *E1La*, and *E1Lb* mutations alternative mutations that cause the same results, or do these mutations show additive effects (like the mutations among *E1*, *E1La*, and *E1Lb*)? Fig. 5h has the necessary information to answer these questions, but Fig. 5h by itself makes it difficult to assess these points. Since *Tof4b* is not the first gene reported to be important for higher latitude adaptation, it is crucial to understand and discuss the effects of other mutations that cause similar effects.

Response: *Tof12*, *E1*, *E1La* and *E1Lb* ultimately regulate flowering time by modulating the expression of the *FT* gene. Whereas, according to previous reports, *E1* is downstream of the *Tof12*, and exerting an epistatic effect on *Tof12*. Typically, fully loss-

of-function mutants are preferred to explore the genetic relationships and effects among genes that may have upstream-downstream associations. Moreover, *e1-wp*, *E1La*, and *E1Lb* are partial loss-of-function mutations rather than complete loss-of-function mutations. Therefore, while we can observe a trend where more accumulation of mutations in these four genes leads to more earlier flowering (Figure was shown below), we hold the opinion that it may strong persuasiveness to assess the relationships and effects of these mutations based on the flowering time of wild soybean populations

Fig Flowering time of a panel of 556 wild soybean.

Question 2: The authors described that E1, E1La, and E1Lb are subfunctionalized. I am not convinced of this statement (hence the title). All three genes are repressors of FT genes. Previous results showed that at least the long-day and short-day expression patterns of all three genes are very similar, and all of them are similarly regulated by phyA (E3/E4). The authors and others (Wang et al., 2022 *Frontiers in Plant Sci*) showed that E1 represses E1La and E1Lb. Could E1 bind to its own promoter to repress its expression (negative feedback)? If not, it could be a new connection, but if E1 already has negative feedback, this mechanism might also be duplicated to E1Ls (which may indicate that E1La or E1Lb could negatively regulate E1 and their

expression as well). The authors need to check the possible connection of E1 repressing E1La and E1Lb as a new pathway by studying the expression of these genes in corresponding lines. At least, based on the information present in this manuscript, there is not enough information to support the statement of subfunctionalization among E1, E1La, and E1Lb.

Response: This is very good suggestion. We appreciate your valuable guidance regarding the functionality of the E1 negatively regulate itself. We conducted additional experiments including Droplet digital PCR, and ChIP-qPCR, transient reporter-expression assays to test this possibility. All these results indicated that E1 could bind its own promoter to repress itself. This self-suppression capability was also found in E1La and E1b. As you suggested, if the E1 gene possessed self-suppression capability before genomic duplication, defining such reciprocal suppression among duplicated E1 family genes as subfunctionalization is inappropriate. So, we decided not to assert gene subfunctionalization at the transcriptional level for the E1 gene family. So we reorganized these data and renamed this part as “The transcriptional regulation among *E1* family”.

Nevertheless, we also observed differential subcellular localization of two E1-like protein compared to E1, resulting in a weakened regulatory capacity on downstream genes. This aligns with the definition of gene subfunctionalization, wherein both copies may become partially compromised by mutation accumulation to the point at which their total capacity is reduced to the level of the single-copy ancestral gene (Lynch & Conery, 2000). Thus, at this level, we still suggest subfunctionalization among the three *E1* genes.

Lynch, M. & Conery, J.S. The evolutionary fate and consequences of duplicate genes. *Science* 290, 1151-1155 (2000)

Question 3: Mainly based on the results shown in Fig. 1, the authors proposed that the 50-bp deletion in the E1Lb promoter region is likely the cause of the *tof4b* phenotype. Could the authors discuss the presence of possible known cis-elements that are possibly important for the induction of E1Lb in the deleted 50-bp?

Response: In response to your comment, we analyzed the promoter elements within

the deleted region and found that it contains a BOX 4 site, two CAAT-boxes site, and three TATA-box sites. Among them, box 4 has been reported as "part of a conserved DNA module involved in light responsiveness"(Yin et al., 2013; Ain-Ali et al., 2021), which is likely to be associated with the observed early flowering phenotype. We have incorporated this content into the discussion section in Line 412-417.

Yin, G., Xu, H., Xiao, S., Qin, Y., Li, Y., Yan, Y., & Hu, Y. (2013). The large soybean (*Glycine max*) WRKY TF family expanded by segmental duplication events and subsequent divergent selection among subgroups. *BMC plant biology*, 13, 148.

Ain-Ali, Q. U., Mushtaq, N., Amir, R., Gul, A., Tahir, M., & Munir, F. (2021). Genome-wide promoter analysis, homology modeling and protein interaction network of Dehydration Responsive Element Binding (DREB) gene family in *Solanum tuberosum*. *PloS one*, 16(12), e0261215.

Question 4: Also, regarding the Fig. 1c result, it would be more informative to understand the possible mechanism of the 50-bp deletion if the authors could analyze the *Tof4b* expression patterns in *Tof4b* and *tof4b* (*tof4b-2*) NIL lines in long days. Please analyze the daily *E1Lb* expression patterns in two NIL lines used for Fig. 1c in long days (like the ones shown in Fig. 2). This result would tell us whether the 50-bp deletion of the promoter reduces the expression of *Tof4b* (*E1Lb*) throughout the day (which means that an unknown activator for *E1Lb* may bind to the 50-bp region the whole day).

Response: We thank you for your insightful comments and agree with your suggestion to add the *Tof4b* expression patterns in NIL lines in Figure 1c. Compared with *NIL-Tof4b*, the promoter of *NIL-tof4b* missing 50 bp, which resulted in the low expression level, suggesting that the deleted 50 bp sequence may contain essential cis-regulatory elements that are crucial for optimal *E1Lb* (*Tof4b*) expression.

Question 5: The e1-wp mutation is found in NC and Russian wild soybeans. If the e1-wp mutation is indeed the weak promoter mutation, the authors should show that *E1* level reduction in the e1-wp containing soybeans. In these lines, what is the expression of *E1La* and *E1Lb*? Are they elevated? Please analyze the expression of *E1*, *E1La*, and *E1Lb* in the e1-wp plants.

Response: We appreciated your comments regarding the analysis on the function of

the weak promoter mutation of *e1-wp*. We have examined the expression levels of the *E1* family genes in ten wild soybeans carrying the *E1* allele and ten carrying the *e1-wp* allele, respectively. As expected, we found differences in the average *E1* expression levels between wild soybeans carrying the *E1* allele and those carrying the *e1-wp* allele. The *E1Lb* expression level in *E1* allele carriers was slightly higher than in those carrying the *e1-wp* allele. While there was no significant difference in the expression levels of *E1La*, which may be attributed to differences in genetic background or the complex interplay of the three *E1* genes. We have incorporated this content into the in Line 377-380 (in results) and Line 504-508 (in discussion).

Supplementary Fig 15 | Expression level of *E1*, *E1La*, and *E1Lb* in wild soybeans with *E1* or *e1-wp* allele. Expression level of *E1* (a), *E1La* (b), and *E1Lb* (c) in ten wild soybeans carrying

E1 allele and ten wild soybeans carrying *e1-wp* allele. ** indicates statistically significant determined by Student's t test ($P < 0.05$).

Question 6: For discussing the effects of amino acid sequence differences in the N-terminal end on the nuclear localization, only showing GFP fluorescence is not that quantitative, as the exposure time difference or expression levels of GFP tagged protein could make the pictures look like nuclear enriched or not enriched as GFP signal by itself already strongly localized in the nucleus. To discuss the changes in nuclear and cytoplasmic distributions of different E1/E1L constructs, the authors should also perform fractionation (nuclear vs cytoplasm) and quantify using western blot analyses.

Response: Thanks for your highly professional suggestions. We have conducted subcellular fractionation experiments, and used western blot analyses to more accurately evaluate the nuclear and cytoplasmic distributions of the different E1/E1L constructs and other experiments related to nuclear localization. The revised figure is presented in Supplementary Fig. 5, 6, 7.

Question 7: What is the *tof12-2* mutation? Fig. 5 shows that the *tof12-2* is enriched in soybeans from higher latitudes. The authors previously showed that *tof12-1* is enriched in soybeans from higher latitudes (Lu et al. Nature Gen 2020). *Tof12* mutations seem important for adapting to higher latitudes, but what is the relationship (allelic distributions) between these two mutations in soybeans from different areas? The authors mention that *tof1-2* has a frameshift mutation but without the data. Please add actual results to support these statements.

Response: We apologized for any confusion caused by the insufficient information provided earlier. *tof12-2* allele is reported to involve a C to T mutation at the 43rd bp of CDS that results in a premature stop codon in previous study (Lu et al. Nature Gen 2020). It is distinct from the *tof12-1* allele (a stop loss mutation in 1879 bp), fixed in cultivated soybeans during domestication in Huanghuai region (Lu et al. Nature Gen 2020). The allele of *tof12-2* is only presents in wild soybeans at high latitudes, and is

completely absent in cultivated soybeans. So, these two alleles were selected through different evolutionary process and in different geographical areas. We have included this information in Line 392-396 of the revised manuscript.

Question 8: For Supplementary Figures 5b, e, and f, it is difficult to assess the strength of each effector construct by looking at the suede color of one transient assay image. The authors should show the averages of measurement values to discuss the strength of each E1/E1L-related effector on the reporter (like the results shown in Supplementary Figures 6c and d).

Response: Thanks for the point. We redo all of the transient reporter-expression assays with 35S: REN to standardize LUC expression.

Question 9: The X-axis level in Fig. 2C has a mistake (this should be the hours of the day). Please correct it.

Response: Sorry for the mistake. We have corrected it.

To Reviewer 3

This is a worthwhile, interesting and wide-ranging study. Ranging from from QTL identification, gene editing, molecular mechanisms to evolutionary and adaptative aspects on three legume specific E1 flowering repressors in soybean with a major focus on E1Lb. These genes have been published on before, but this work lifts understanding of function and impact of the genes in soybean adaptation and cultivation.

Response: We would like to thank the reviewer for the positive response to our manuscript. Below we have addressed the points the reviewer has raised.

Question 1: 68. adaptation is &

Question 2: 70. indicate that E2 is a soybean GIGANTEA

Question 3:FRUITFULL

Question 4: 83. how Tof4b

Response to Question 1-4: We apologized for all the mistakes. We have corrected them in accordance with the advice.

Question 5:104. Explain that plants with A sequences ie from DN50, in region of interest, flower earlier than plants with B ie W82.

Response: Upon reaching this stage of experimentation, we believe that this chromosomal region harbors a gene controlling soybean flowering time. Plants carrying the DN50 type (A) chromosomal segment carry the early-flowering allele variation of this gene. Conversely, plants harboring the Wm82 (B) chromosomal segment carry the late-flowering allele variation of this gene. Subsequent investigations have revealed that this gene corresponds to *E1Lb*.

Question 6: 108 . Explain which of the parents have the different mutations in E1Lb promoter ie . in text it indicates that the DN50 parent would have the deletion mutation.

Response: We apologized for the lack of information provided. We have corrected it as "The E1Lb coding sequence was the same in Wm82 and DN50, but three variants

were found in the E1Lb promoter of the parents: a 50-bp deletion and two SNPs in DN50 (Fig 1b).”

Question 7: 108. The text and figure 1 do not agree. On Fig 1, the W82 parent has the deletion mutation? These needs correcting.

Response: We apologized for the mistake. We have corrected it.

Question 8: 147 Add reference for statement about E1 and E1La binding the promoters of FT2a and FT5a.

Response: Thank you for raising this. We have added the reference (Dong, L. et al. The genetic basis of high-latitude adaptation in wild soybean, 2023, Curr Biol 33, 252-262.) in Line 153.

Question 9: 147. The text and figure 2 do not agree. On figure it is P1 and P4 fragments for both FT genes, in text it is P2 and P4 for FT2a. This needs correcting.

Response: This was a mistake, and we are really sorry for that. We have changed it to “Tof4b–Flag directly binds to the P1 and P4 fragments in the *FT2a* promoter and the P1 and P4 fragments in the *FT5a* promoter (Fig. 2e–h)”.

Question 10: 181. The text and figure Supplementary 5b do not agree. Constructs 4+5 top right have more luminescence than 2+5 bottom right. This implies that the shorter E1 is a stronger repressor than the longer E1. This needs correcting.

Response: We apologized deeply for our oversight. In accordance with the suggestion from other reviewers, we have redesigned this section of experiments, and the relevant details are now presented in Supplementary Figure 7e.

Question 11: 214. The text and figure 3e do not agree. It should read that ... expression levels of....are much higher thanThis needs correcting.

Question 12: 215. The text and Supplementary Figure 5 are incorrectly referred to. Should be Figure 5g,h.

Question 13: 244. Fig 5a needs correcting to Fig 4a.

Response to Question 11 and 13: We apologize for the mistakes. We have changed them.

Question 14: 251. Phrase S53R differs ...seems out of place here.

Response: Thanks for the point. We have changed it as "H4 has a base substitution at the 159th position of the coding region, causing a substitution from Serine (S) to Arginine (R) at residue 53, while the 53rd amino acid, R, differs from those of other E1 homologues (Supplementary Fig. 10a)." in line 274-277.

Question 15: 264 ..of wild andneed to insert missing adjective

Question 16: 458. intensity of umol...not mmol

Question 17: 511. qPCR not qRT-PCR

Response to Question 15 and 17: We are sorry for the writing errors. We all amended them.

Question 18: Fig 1a. – Indicate what the units are for flowering time in the graph. If Dae indicate what this stands for how flowering measured.

Response: We apologize for the lack of information provided. DAE means days after emergence in this manuscript. We have defined it in Fig. 1.

Question 19: Fig 1e. Seems unusual that the promoter LUC fusions eg in Fig 1e and others, without any effectors co transfected, can give LUC expression in Benth leaves at the same level as the highly expressed 35S :REN? Please comment.

Response: Thank you for your attention to detail. We redo all of the transient reporter-expression assays with 35S: REN to standardize LUC expression.

Question 20: Fig 1b. As noted above, error in that DN50 has the deletion allele not W82.

Response: Thank you for the reviewer's suggestion regarding the annotation of Fig.

1b.

Question 21: Fig 1. Define **

Response: We have added the definition for **.

*** indicates statistically significant differences determined by Student's test ($P < 0.01$)”

Question 22: Fig 3. Remove – different letters indicate. These letters are not shown.

Response: We apologize for the mistake. We have removed them.

Question 23: Fig 4. Table in Fig 4a needs correcting and clarifying. Label the last columns. Also AA identified in W82 needs to swapped with AA# in W82. Define fs.

Response: We have changed it accordingly.

Question 24: Figure 6. Too small font. Define gray and blue in right hand panels.

Response: We sincerely apologize for the oversight in our formatting. We have made adjustments by relocating the images from the right portion to Supplementary Figure 15 and rewrite figure legend.” Gray panels represent loss of function allele carriers, and blue panels represent functional type of allele carriers.”

Question 25: Suppl fig 1. Co-transfection assay- but no effector is added- is that right?

Again it is unusual that it is expressed strongly, especially the W82 promoter.

Response: We apologize for the mistake. We have changed it accordingly.

Question 26: Suppl fig 1 The figure legend and the diagram to not agree. Diagram has different constructs that the legend? Diagram has pE1La promoter , while in legend it is proE1Lb (W82)? This needs correcting.

Response: Thank you so much for your careful check. We have changed them (*pE1La* was changed to *pE1Lb*).

Question 27: Suppl fig 3. Define DAE

Response: We apologize for the lack of information provided. DAE means days after emergence in this manuscript. We have defined it in Supplementary Fig. 3.

Question 28: Suppl fig 5. Text and figure did not agree as above- Check in b that construct combinations correctly assigned.

Response: Thanks for your help. We feel really sorry for our carelessness. We have changed them.

Question 29: Suppl fig 5b, e, f. Reporter should be pFT2a not pPT2a.

Response: This was a mistake, and we are really sorry for that. We have changed them.

Question 30: Suppl fig 5. As above comment on why strong expression of control FT2a promoter LUC without effectors in Benth?

Response: Because we did not use the 35S:REN to standardize LUC expression in early experiment, the result in Suppl fig 5 unable to clarify the intensity. This time, we redo all of the transient reporter-expression assays with 35S:REN to standardize LUC expression.

Question 31: Suppl fig 6b. v small font for Tof alleles.

Response: Thanks for this point. We have changed it.

Question 32: Suppl Fig 8. Explain in fig legend and text why there are two Group 1 promoters and two group 2 promoters assayed in the figure. Are they the same or different from each other? What is HY107?. Needs explaining.

Response: The figure showed the nucleotide polymorphisms of the gene region of *E1* gene. Abundant nucleotide polymorphisms distinctly divide the schematic into two groups. Upon closer examination, we find that 6 SNPs and 4 InDels segregate the major cluster into two subgroups. In fact, the E1 promoters of individuals within each group are not entirely identical either. Therefore, we selected two promoters with slight

differences from each group to test their activity. HY107 is vector of transient-reporter assays. We revised the schematic for the avoidance of confusion.

Question 33: Suppl fig 9. Legend needs correcting. States: .. cultivated soybean in 851 wild soybeans. Which is it, cultivated or wild?

Response: We have reworded this figure legend to make it more clarity.

To Reviewer 4

Fang et al present a dense and comprehensive investigation of the soybean E1 gene family and target genes and alleles of those genes controlling adaptation to high latitudes. There are a multitude of new discoveries in this work that exemplify deep exploration of the genetic systems responsible for this important phenotype. Generally, I found the work scientifically suitable for publication in Nature Communications with one request for an expanded discussion for the possible authentic start site for the E1 gene (currently described without its own results heading and in Supplementary Figure 5).

Response: Thank you very much for your insights and suggestions. To further identify the true initiation site of the *E1* gene, we changed the two start codons from ATG to ATT individually, and found that both of the two proteins could be detected. This result indicated that *E1* possesses two translation initiation sites (Supplementary Fig. 5). Using transient-reporter assays, we found that the longer E1 protein has a much stronger transcriptional inhibitory activity on the promoter of FT2a than the previously considered E1, suggesting that the longer E1 protein possesses greater biological activity.

Question 1: Please refer to the gene IDs (Wm82.a2.v1 or other) at least once for all of the genes presented in the manuscript to avoid confusion of different naming schemes (for example, Tof4 [Glyma.04g156400]).

Response: Thank you for the point. We have added the gene IDs for all of the genes in the full text.

Question 2: Line 68 “in” should be” is”

Response: We are sorry for the writing errors. We have changed it.

Question 3: Line 95-96 is not complete sentence: “While Tof4 alone cannot fully explain the flowering differences caused by this interval.”

Response: We have changed the sentence and add one reference in line 99 to

“However, *Tof4/E1la* alone cannot fully explain the flowering differences caused by this chromosomal region”.

Question 4: Line 96-99 This sentence is also a little confusing and could be reworded.

Response: We have changed the sentence in line 100-103 to “Further inspection revealed a secondary population with homozygosity at the interval including E1La, but heterozygosity for the interval between 15,526,145 bp to 29,675,002 bp showed a separation of flowering time. These suggest the existing of another locus (*Tof4b*)”.

Question 5: Line 131 This section-what tissue was characterized? Missing from M&M.

Response: We apologize for the lack of information provided. In this section, we use trifoliolate leaves as samples to quantify the expression of relevant genes. We have added this information in M&M.

Question 6: Lines 155-173 Promoting Supp. Figure 4 to the main Results section might help with interpretation of the associated text.

Response: In our initial plan, we also considered placing this figure in the main text. However, with the addition of other related images, the page lacks the capacity to accommodate the inclusion of this picture.

Question 7: Line 184-191 Complete sentence? Please reword.

Response: We have rephrased these sentences to make it more clarity and conciseness.

“Whereas, the E1La-eGFP and E1Lb-eGFP proteins were distributed in both the nucleus and the cytoplasm, exhibits a distribution pattern similar to that of e1-as protein whose nuclear localization signal region has undergone mutation.”

Question 8: Line 268 For Figure 4d, it was too small for me to see properly.

Response: Thank you for your suggestion. In order to make the image clear, we have moved it to Supplementary Figure 11.

Question 9: Line 413 Blue text?

Response: We are sorry for the oversight. We have now corrected it to black font color.

Question 10: Line 429 Please reword for clarity.

Response: Sorry for the unclear sentence. We have reworded these sentences to make them more clarity and conciseness.

Question 11: Line 433. Supp Figure 9 legend is unclear for last sentence. a is wild soybean and b is cultivated? The legend states "Data from cultivated soybeans in an ..."

Response: We have reworded this figure legend to make it clearer.

Reviewers' Comments:

Reviewer #1:

Remarks to the Author:

The authors have adequately addressed my previous questions. It is a great paper!

Reviewer #2:

Remarks to the Author:

In this revised manuscript, the authors responded to all of my previous comments. New results and analysis undoubtedly strengthened their statements. Most new analyses were adequately completed, and I don't have further comments on the contents. I noticed a couple of minor issues/mistakes with the new data, and I would like the authors to fix them before publication.

1. Regarding my previous question 1, the authors added Supplementary Fig 17, depicting the possible linkage with four natural early flowering mutations (e1-wp, tof4-2, tof4b-1&3, and tof12) in the wild soybean population. This figure requires a proper figure legend. For example, please explain what population was used to make this figure. Based on this, is the tof4-2 mutation linked to the tof12?
2. There are typos in the labeling in Supplementary Fig 7c. "GFP" on the left panel should be "E1-12-GFP," and "E1-12-GFP" on the right panel should be "E1-13-GFP."

Reviewer #3:

Remarks to the Author:

1. The authors have carried out additional work which is good to see and have fixed some of the mistakes in figures and text.
2. Unfortunately, there are still some of the same errors and new ones are evident in this version of the manuscript. Also some misconceptions are evident.
3. 340 Appears to be a major error? ...text states: least functional allele tof4b-2 is enriched in extremely high latitudes ...But the problem is you do not show it on Figure 5a which is cultivated soybean distribution. I.e. Only b-1 and b-3 are shown according to key.
4. Similarly Fig 5b refers to wild soybean. But b-2 allele data is not shown on the graphs? Please explain.
5. 180 Misconception between transcription and translation. Refers to potential transcription initiation site ahead..., but there are no transcription initiation sites shown on the figure, but there is a potential new translation site (ATG). Transcription and translation initiate from different positions. I think therefore you mean a new potential translation initiation site and this needs correcting.
6. 183 Similarly, text says both transcripts are indeed present (Suppl Fig 5b-d). But this figure does not show transcripts, it shows GFP translational fusions and their location. Needs correcting. Both proteins are indeed present...
7. 413 and other parts of text. Refer to tof1b-2 as a loss of function. It has a 50 bp deletion in promoter. It is still expressed in fig 1c, what is other evidence that it is a true loss of function allele?
8. Suppl figure 5c The alignment of the PEPC and H3 and GFP blots are not correct with the T, C and N lane names. This is because there is a ladder showing in the GFP lane 1, ie it is not Total protein in the first lane of GFP. This all needs realigning and correcting.
9. 201. The replacement of the N-terminus of E1a and b with E1 N terminus- the notation is very confusing in text and on the Figure 3b. The triangle symbol is used for the N terminus of all 3 proteins so does not distinguish properly between the different proteins N terminus.
10. Therefore, I suggest in both places call it instead E1triangle-E1a and E1 triangle E1b. And better still the triangle sign could be replaced with another symbol in figure and text. This is because the triangle sign also is used for delta, which means a deletion, so it is confusing. Please clarify.
11. 238: Transient reporter assay is indicated as in Fig 3g-i. But Fig 3g-i does not show a transient

- reporter assay. It is a ChIP qPCR. Needs correcting.
12. Fig 3 f. the legend says schematic of constructs for reporter..but this is wrong. It is a DD PCR. Needs correcting.
 13. 240 refer to Fig 3h-I, but it is Fig3 g-i. Needs correcting.
 14. 107: Remove the description in text of of A and B sequence- because the figure no longer has A and B sequence on it.
 15. Fig 1a. – In figure, Put the units for flowering time on the graph axis ie Flowering time (DAE). Remove Flowering time variations
 16. Fig 1a In figure key replace heterogeneious with Heterozygous
 17. Fig 1a in Legend, remove the text referring to A and B sequence because there is no A and B on figure anymore.
 18. Figure 1a: Indicate in legend what the growth conditions were for Figure 1a, they are missing at present
 19. 154 refer to Fig 2e-h, but there is no g and h in this figure, so its 2e,f
 20. Suppl Fig 1a the colours in key are confusing. Suggest remove the key and label the current 5'UTR and 3'utr on the figure.
 21. Suppl figure 5c. Please indicate in legend what T, N and C stand for. Please add.
 22. Suppl figure 5c. Indicate in legend what the controls PEPC and H3 are used for and what they stand for. Please add.
 23. Suppl figure 5d Indicate how the image intensity was measured in legend and in the methods sections. Please add.
 24. Suppl figure 6 – It is called Fig 6 at present- but it's in the supplementary figure sections, so this needs correcting.
 25. Suppl figure 6 - In legend indicate how intensity measured for c. Please add.
 26. 255. the sentence is not finished properly among Please correct.
 27. 267 Section on evolutionary trajectory. There are some big problems with clarity here and errors.
 28. Fig. 4a table and legend is still not clear.
 29. Fig. 4a Wn82 should Wm82 throughout,
 30. Fig. 4a AA identified in Wm82 – unclear what the last 2 columns show us, but this appears wrong as they are both numbers not AA
 31. Fig. 4a AA # in Wn82_ both columns show AAs not numbers so this is wrong and needs correcting.
 32. Fig. 4a AA change spelling should be AA change.
 33. Fig. 4a Tell us what fs stands for in the legend
 34. Fig. 4a Please give the last two columns a title
 35. Fig. 4a Describe in legend that the blue cells in table refer to the 50 bp deletion in the promoter.
 36. Fig 4c – explain why tof4b-3 is not shown?
 37. 271 tof4b-2 also known as H3. Please specify the frameshift mutation- where specifically does it affect the predicted protein?
 38. 280 In suppl fig 10, the old notation is still used on this figure ie H notation rather than b alleles. This needs correcting in text, or on figure.
 39. 300, 301, 304 - why are b-1 b-2 considered together? Because b-2 can be differentiated based on the frameshift? Please comment.
 40. 325 Wrong figure referred to; should be Fig 13a,b, not fig 14. Please correct.
 41. 350 _ refer to Suppl fig 13, not Fig 14.
 42. Fig 5- insert (a) into legend as its missing.
 43. Fig 5b refers to wild soybean. But b-2 allele data is still not shown on the graphs??
 44. 368 need to add in fig 5c as is missing currently from text.
 45. 394 Clarify what is a stop loss mutation? Do you mean a premature stop codon?
 46. 413 50 bp deletion- examine elements that might be important for its function. This needs to come in the results section, not in discussion.
 47. 440 Interestingly, the introduction ...This is repetitive of information above
 48. 495 states tof4 is localized close to tof4b Chromosomes 04? This needs clarifying.
 49. 504 refers to suppl Fig 17. Suppl fig 17 has no legend. A legend needs to be added to describe what we are looking at and how it was obtained.

50. Methods. No Crispr Cas9 gene editing method and plant analysis described. This needs to be added.
51. 624 Add a bit more detail about DAPI staining and detection method in N benth.
52. 650 Indicate how protein intensity was measured?

Reviewer #4:

Remarks to the Author:

This is a review of the revised manuscript. The authors expanded their already comprehensive analysis of the soybean E1 family of genes. I found the new experiments on the expanded aspects of the translation initiation, effects of the different fused E1 (delta) and mutated start codons, as well as the effect of the T12K amino acid divergence for E1 and E1LA or E1LB experiments to add fill an important knowledge gap.

There was an alignment issue in Supplemental Figure 5c such that the MW marker is in the 1T position instead of the experimental lane. Also, there is an issue in the legend for that figure-(f) should be 9e). Finally, the revised text could be improved by an English language edit throughout the manuscript. In some of the revised text, there were incomplete sentences or sentences that were very awkwardly worded.

Point-by-point responses to reviewers' comments

Reviewer #1 (Remarks to the Author):

The authors have adequately addressed my previous questions. It is a great paper!

We wish to extend our heartfelt thanks for your valuable contributions and suggestions in enhancing this paper.

Reviewer #2 (Remarks to the Author):

In this revised manuscript, the authors responded to all of my previous comments. New results and analysis undoubtedly strengthened their statements. Most new analyses were adequately completed, and I don't have further comments on the contents. I noticed a couple of minor issues/mistakes with the new data, and I would like the authors to fix them before publication.

We would like to express our appreciation for your assistance and valuable suggestions on this paper and have sought to address these remaining points below.

1. Regarding my previous question 1, the authors added Supplementary Fig 17, depicting the possible linkage with four natural early flowering mutations (*e1-wp*, *tof4-2*, *tof4b-1&3*, and *tof12*) in the wild soybean population. This figure requires a proper figure legend. For example, please explain what population was used to make this figure. Based on this, is the *tof4-2* mutation linked to the *tof12*?

Response: 75% (120/160) of *tof12-2* alleles are in the *tof4-2* background, implying that *tof12-2* might have arisen after *tof4-2*. Considering that *Tof12* and *Tof4* are not on the same chromosome, there is not enough evidence here to conclude the *tof4-2* mutation is linked to the *tof12-2* mutation. We have adjusted the figure legend accordingly for **Supplementary Fig. S17**.

2. There are typos in the labeling in Supplementary Fig7c. “GFP” on the left panel should be “E1-12-GFP,” and “E1-12-GFP” on the right panel should be “E1-13-GFP.”

Response: We have revised the labelling for this panel.

Reviewer #3 (Remarks to the Author):

1. The authors have carried out additional work which is good to see and have fixed some of the mistakes in figures and text.

2. Unfortunately, there are still some of the same errors and new ones are evident in this version of the manuscript. Also some misconceptions are evident.

Response: Thank you for kindly drawing our attention to these remaining inconsistencies, which we trust we have now fully resolved.

3. 340 Appears to be a major error?text states: least functional allele *tof4b-2* is enriched in extremely high latitudesBut the problem is you do not show it on Figure 5a which is cultivated soybean distribution. Ie. Only b-1 and b-3 are shown according to key.

Response: We apologise for this oversight. **Fig. 5a** shows the global distribution of *tof4b* alleles present in cultivated soybeans. *tof4b-3* is only found in wild soybeans and not included therein. Therefore, dark blue should represent *tof4b-2* in **Fig. 5a**. We have now revised the figure key accordingly.

4. Similarly Fig 5b refers to wild soybean. But b-2 allele data is not shown on the graphs? Please explain.

Response: In **Fig. 5b**, we present only the distribution of *Tof4b* alleles in wild soybeans. *tof4b-2* is only found in cultivated soybean (**Fig. 4b**), so the *tof4b-2* allele data are not shown in this panel.

5. 180 Misconception between transcription and translation. Refers to potential transcription initiation site ahead..., but there are no transcription initiation sites shown on the figure, but there is a potential new translation site (ATG). Transcription and translation initiate from different positions. I think therefore you mean a new potential translation initiation site and this needs correcting.

Response to Question 5: We apologize for this confusion. It should indeed be the **translation** initiation site and we have revised the associated wording accordingly.

6. 183 Similarly, text says both transcripts are indeed present (Suppl Fig 5b-d). But this figure does not show transcripts, it shows GFP translational fusions and their location. Needs correcting. Both proteins are indeed present...

Response: We have change it to “we found that both proteins are indeed expressed and do not markedly affect their corresponding nuclear localisation.” in **Line 184**.

7. 413 and other parts of text. Refer to *tof1b-2* as a loss of function. It has a 50 bp deletion in promoter. It is still expressed in fig 1c, what is other evidence that it is a true loss of function allele?

Response: Based on your suggestion, we have modified the description of the allelic variations of *tof4b*. We now describe *tof4b-1* carrying a 50-bp deletion in the promoter as weak allele, *tof4b-2* carrying a 50-bp deletion and a frameshift mutation as loss-of-function allele, and *tof4b-3* carrying an amino-acid substitution as weak allele. We have also modified the relevant descriptions in the revised manuscript.

8. Suppl figure 5c The alignment of the PEPC and H3 and GFP blots are not correct with the T, C and N lane names. This is because there is a ladder showing in the GFP lane 1, ie it is not Total protein in the first lane of GFP. This all needs realigning and correcting.

Response: We have relabelled these images.

9. 201. The replacement of the N-terminus of E11a and b with E1 N terminus- the notation is very confusing in text and on the Figure 3b. The triangle symbol is used for the N terminus of all 3 proteins so does not distinguish properly between the different proteins N terminus.

10. Therefore, I suggest in both places call it instead E1triangle-E11a and E1 triangle E11b. And better still the triangle sign could be replaced with another symbol in figure and text. This is because the triangle sign also is used for delta, which means a deletion, so it is confusing. Please clarify.

Responses: We have modified the descriptions to make it easier to understand and replaced Δ with Ω .

11. 238: Transient reporter assay is indicated as in Fig 3g-i. But Fig 3g-i does not show a transient reporter assay. It is a ChIP qPCR. Needs correcting.

Response: Corrected.

12. Fig 3 f. the legend says schematic of constructs for reporter..but this is wrong. It is a DD PCR. Needs correcting.

Response: Corrected.

13. 240 refer to Fig 3h-I, but it is Fig3 g-i. Needs correcting.

Response: We performed ChIP–qPCR to test whether E1 physically associates with the promoter of *E1* (**Fig. 3g**), *E1La* (**Fig. 3h**), *E1Lb* (**Fig. 3i**), so we prefer to retain **Fig. 3g–i** here to call out these experiments.

14. 107: Remove the description in text of of A and B sequence- because the figure no longer has A and B sequence on it.

Response: Corrected.

15. Fig 1a. – In figure, Put the units for flowering time on the graph axis ie Flowering time (DAE). Remove Flowering time variations

Response: Corrected.

16. Fig 1a In figure key replace heterogeneious with Heterozygous

Response: Corrected.

17. Fig 1a in Legend, remove the text referring to A and B sequence because there is no A and B on figure anymore.

Responses: Corrected.

18. Figure 1a: Indicate in legend what the growth conditions were for Figure 1a, they are missing at present.

Responses: Corrected.

19. 154 refer to Fig 2e-h, but there is no g and h in this figure, so its 2e,f

Response: Corrected.

20. Suppl Fig 1a the colours in key are confusing. Suggest remove the key and label the current 5'UTR and 3'utr on the figure.

Response: Corrected.

21. Suppl figure 5c. Please indicate in legend what T, N and C stand for. Please add.

22. Suppl figure 5c. Indicate in legend what the controls PEPC and H3 are used for and what they stand for. Please add.

Responses: We have incorporated the relevant information into the legend for **Fig. S5C:**

(c) Subcellular fractionation analysis of various proteins as summarised in (b). *N. benthamiana* leaves were left to transiently express for 48–72 h after infiltration the indicated constructs before

harvest and protein extraction. E1–GFP isoforms were detected using anti-GFP antibodies, PEPC was detected with anti-PEPC antibodies as the cytoplasmic-fraction marker and H3 proteins were detected using anti-H3 antibodies as the nuclear-fraction marker. T: Total, C: Cytoplasmic fraction, N: Nuclear fraction. 23. Suppl figure 5d Indicate how the image intensity was measured in legend and in the methods sections. Please add.

Response: All immunoblot bands were analysed for band grayscale values using ImageJ. We have now provided the relevant information in the respective figure legends and methods & methods sections.

24. Suppl figure 6 – It is called Fig 6 at present- but it's in the supplementary figure sections, so this needs correcting.

Response: Corrected.

25. Suppl figure 6 - In legend indicate how intensity measured for c. Please add.

Response: Corrected, as per point 23 above.

26. 255. the sentence is not finished properly among Please correct.

Response: Corrected.

27. 267 Section on evolutionary trajectory. There are some big problems with clarity here and errors.

Response: We have made substantial textual modifications to this section.

28. Fig. 4a table and legend is still not clear.

29. Fig. 4a Wn82 should Wm82 throughout,

30. Fig. 4a AA identified in Wm82 – unclear what the last 2 columns show us, but this appears wrong as they are both numbers not AA

31. Fig. 4a AA # in Wn82_ both columns show AAs not numbers so this is wrong and needs correcting.

32. Fig. 4a AA change spelling should be AA change.

33. Fig. 4a Tell us what fs stands for in the legend

34. Fig. 4a Please give the last two columns a title

35. Fig. 4a Describe in legend that the blue cells in table refer to the 50 bp deletion in the promoter.

Responses to 28–35: We have revised **Fig. 4a** and modified its legend according to these suggestions.

36. Fig 4c – explain why *tof4b-3* is not shown?

Response: *tof4b-3* should have been shown in **Fig. 4c**. We have now corrected it.

37. 271 *tof4b-2* also known as H3. Please specify the frameshift mutation- where specifically does it affect the predicted protein?

Response: The *tof4b-2* mutation results in a truncation after 73 amino acids in the 192-amino-acid *Tof4b* protein. We have provided this information in **Lines 270–271**.

38. 280 In suppl fig 10, the old notation is still used on this figure ie H notation rather than b alleles. This needs correcting in text, or on figure.

Response: Corrected throughout.

39. 300, 301, 304 - why are b-1 b-2 considered together? Because b-2 can be differentiated based on the frameshift? Please comment.

Response: When discussing the introgression of genomic fragments, we refer to the introgression of genomic fragments carrying the 50-bp deletion from the *Tof4b* promoter. *tof4b-2* not only has a frameshift mutation, but also has the 50-bp deletion.

Therefore, when analysing the size of the introgression fragment by calculating genomic differentiation (F_{ST}) between accessions carrying or lacking the deletion, we also incorporated *tof4b-2* into accessions carrying the 50-bp deletion.

40. 325 Wrong figure referred to; should be Fig 13a,b, not fig 14. Please correct.

Response: Corrected.

41. 350 _ refer to Suppl fig 13, not Fig 14.

Response: Corrected.

42. Fig 5- insert (a) into legend as its missing.

Response: Corrected.

43. Fig 5b refers to wild soybean. But b-2 allele data is still not shown on the graphs??

Response: *tof4b-2* only exists in cultivated soybeans, so it is not shown in **Fig. 5b**.

44. 368 need to add in fig 5c as is missing currently from text.

Response: Corrected (**Fig. 5c–e**).

45. 394 Clarify what is a stop loss mutation? Do you mean a premature stop codon?

Response: The *tof12-1* allele has a stop-gain mutation, leading to the introduction/gain of a premature stop codon. We have corrected this throughout.

46. 413 50 bp deletion- examine elements that might be important for its function. This needs to come in the results section, not in discussion.

Response: We have moved this part to the results (**Lines 111–113**).

47. 440 Interestingly, the introduction ... This is repetitive of information above

Response: We have deleted this sentence.

48. 495 states *tof4* is localized close to *tof4b* Chromosomes 04? This needs clarifying.

Response: We apologise for this confusion. This sentence now reads “Among them, *Tof4* is located close to *Tof4b* on chromosome 4” in **Line 483**.

49. 504 refers to suppl Fig 17. Suppl fig 17 has no legend. A legend needs to be added to describe what we are looking at and how it was obtained.

Response: Corrected (**Supplementary Fig. S17**).

50. Methods. No Crispr Cas9 gene editing method and plant analysis described. This needs to be added.

Response: Methodologies have now been added.

51. 624 Add a bit more detail about DAPI staining and detection method in *N. benthamiana*.

Response: We have incorporated the following information into the methods:

DAPI was used to stain nuclei in 1× PBS at a working concentration of 100 ng/mL. *N. benthamiana* leaf samples were placed on a microscope slide to which a few drops of DAPI staining solution were and allowed to incubate for 10 min prior to imaging. Samples were observed under a confocal laser-scanning microscope with an excitation wavelength of 360–400 nm and emission wavelength of 460–500 nm.

52. 650 Indicate how protein intensity was measured?

Response: We have now incorporated this into the ‘Cell-fractionation assays’ methods section.

Lastly, we want to reiterate our appreciation for your assistance and suggestions on this manuscript.

Reviewer #4 (Remarks to the Author):

This is a review of the revised manuscript. The authors expanded their already comprehensive analysis of the soybean E1 family of genes. I found the new experiments on the expanded aspects of the translation initiation, effects of the different fused E1 (delta) and mutated start codons, as well as the effect of the T12K amino acid divergence for E1 and E1LA or E1LB experiments to add fill an important knowledge gap.

There was an alignment issue in Supplemental Figure 5c such that the MW marker is in the 1T position instead of the experimental lane. Also, there is an issue in the legend for that figure-(f) should be 9e).

Finally, the revised text could be improved by an English language edit throughout the manuscript. In some of the revised text, there were incomplete sentences or sentences that were very awkwardly worded.

Response: Thank you for pointing these out. We have rearranged **Fig. 5c**, updated the callouts, and received language support. We would like to express our gratitude for your valuable suggestions on this paper.

Reviewers' Comments:

Reviewer #3:

Remarks to the Author:

The paper is looking very good overall.

I noticed the following that need correcting.

1. This 107 region harbours a member of the E1 family, E1Lb (Glyma.04G143300), a floral 108 repressor controlling night-break responses that is associated with early flowering under non-inductive long-day photoperiods^{26,29}

-Early flowering should be late flowering. Please correct.

2. 181 . We found a potential transcription-initiation site upstream of the 182 previously annotated translation initiation site, extending the length of E1 to 193 aa,

-Transcription initiation should be translation initiation. Please correct.

3. 875 (d) Schematic of the constructs used for a transient reporter-gene assay to validate the 876 casual mutations of E1Lb.

-Casual should be causal. Please correct.

Dear Reviewers,

Thanks again for your constructive suggestions to our manuscript.

1. This 107 region harbours a member of the E1 family, E1Lb (Glyma.04G143300), a floral repressor controlling night-break responses that is associated with early flowering under non-inductive long-day photoperiods. -Early flowering should be late flowering. Please correct.
2. We found a potential transcription-initiation site upstream of the previously annotated translation initiation site, extending the length of E1 to 193 aa. - Transcription initiation should be translation initiation. Please correct.
3. 875 (d) Schematic of the constructs used for a transient reporter-gene assay to validate the 876 casual mutations of E1Lb. -Casual should be causal. Please correct.

Response to 1-3: We are so sorry for these mistakes. We have changed them.